# Ribonuclease 4 is associated with aggressiveness and progression of prostate cancer

Nil Vanli[1,2], Jinghao Sheng[1,3], Shuping Li[1], Zhengping Xu [3] & Guo-Fu Hu [1,2 ✉]

Prostate specific antigen screening has resulted in a decrease in prostate cancer-related deaths. However, it also has led to over-treatment affecting the quality of life of many patients. New biomarkers are needed to distinguish prostate cancer from benign prostate hyperplasia (BPH) and to predict aggressiveness of the disease. Here, we report that ribonuclease 4 (RNASE4) serves as such a biomarker as well as a therapeutic target. RNASE4 protein level in the plasma is elevated in prostate cancer patients and is positively correlated with disease stage, grade, and Gleason score. Plasma RNASE4 level can be used to predict biopsy outcome and to enhance diagnosis accuracy. RNASE4 protein in prostate cancer tissues is enhanced and can differentiate prostate cancer and BPH. RNASE4 stimulates prostate cancer cell proliferation, induces tumor angiogenesis, and activates receptor tyrosine kinase AXL as well as AKT and S6K. An RNASE4-specific monoclonal antibody inhibits the growth of xenograft human prostate cancer cell tumors in athymic mice.

[1] Divison of Hematology and Oncology, Department of Medicine, Tufts Medical Center, Boston, MA, USA. [2] Graduate Program in Biochemistry, Graduate School of Biomedical Sciences, Tufts University, Boston, MA, USA. [3] Institute of Environmental Medicine, Zhejiang University School of Medicine, Hangzhou, China. ✉email: guo-fu.hu@tufts.edu

Prostate cancer is common in American men. The incidence rate is expected to grow with the aging population[1]. The most commonly used diagnosis tools in prostate cancer are prostate specific antigen (PSA) test, digital rectal exam, ultrasound in combination with guided biopsy and histopathological Gleason grading[2]. Of these, only PSA can be classified as a biomarker but is limited[3,4] in predicting prognosis, which leads to either over-treatment or under-treatment[5]. Furthermore, PSA is not a biomarker specific for prostate cancer as its level is also elevated in prostatitis and in benign prostatic hyperplasia (BPH)[6]. Several other biomarkers have been discovered such as PSA derivatives and prostate cancer antigen 3[7]. However, multiple studies have failed to demonstrate independent values of these markers to predict cancer progression[8,9]. It is thus necessary to identify noninvasive biomarkers that can provide a higher degree of specificity for detecting aggressiveness and for predicting progression of prostate cancer. There is also unmet clinical needs for alternative therapies for prostate cancer, especially for patients with advanced or aggressive cancers. Prostate cancer deaths are typically the result of aggressive metastases that are unresponsive to androgen deprivation therapy[10].

Human pancreatic ribonucleases (RNases) are a large superfamily with diverse functions[11] including growth and survival, angiogenesis, neurogenesis, immune-regulation, and host defense[12–18]. Among the 13 members of this superfamily, ribonuclease 4 (RNASE4) is unique with two distinct features. It has the most conserved amino acid sequence among vertebrate species (94% identity among human, bovine, mouse, and porcine)[19–22], and has the strictest nucleotide specificity as it only cleaves at the 3' side of uridine[23,24]. These features suggest that RNASE4 has a unique biological role.

In this study, we demonstrate that RNASE4 is up-regulated in prostate cancer, and has diagnostic, prognostic and therapeutic values. We report that RNASE4 expression is increased progressively in prostate cancer, correlates with aggressiveness of the disease, can accurately distinguish between healthy, BPH, and prostate cancer, and can independently predict biopsy outcome. In addition, we uncovered a previously unknown mechanism of action in which RNASE4 mediates prostate cancer cell proliferation and tumor growth by activating AXL receptor tyrosine kinase and downstream effectors AKT and S6. We also demonstrate that RNASE4 inhibition decreases prostate cancer growth in vitro and in vivo, underscoring a therapeutic potential of RNASE4 inhibitors in prostate cancer treatment.

## Results

**Plasma RNASE4 level is elevated in prostate cancer patients.** RNASE4 level in the plasma samples of healthy control subjects ($n = 120$) and prostate cancer patients ($n = 120$) were determined by an in-house prepared ELISA. The demographics and clinical characteristics of the study population are shown in Supplementary Table 1. There was no correlation between RNASE4 amount and patient demographics, such as age and race (Supplementary Table 2). The intra- and inter-assay precision of the in-house ELISA were shown to be 95.1–98.2% and 90.7–94.1%, respectively (Supplementary Table 3). The specificity and sensitivity of both pAb and mAb of RNASE4, as well as a representative ELISA standard curve were shown in Supplementary Fig. 1. The recovery of RNASE4 spiked to levels throughout the range of the assay in plasma matric was also evaluated and shown to have a recovery range of 113–126% (Supplementary Table 4). The median RNASE4 amount in plasma of prostate cancer patients ($155.4 \pm 2.8$ ng/ml) were higher ($p < 0.0001$) than in control subjects ($101.5 \pm 1.9$ ng/ml) (Fig. 1a). The predictive value of RNASE4 was explored using ROC curve analysis (Fig. 1b),

which shows an AUC of 0.94 (0.91–0.97) at 95% CI. The diagnostic accuracy of RNASE4 depended on the cut-off values (Supplementary Table 5). At the optimal cut-off value of 117 ng/ml, RNASE4 has a diagnostic accuracy of 86%, sensitivity of 94%, specificity of 80%, positive predictive value (+PV) of 83%, negative predictive value (-PV) of 93%, positive likelihood ratio (+LR) of 4.71 and negative likelihood ratio (-LR) of 0.07 (Fig. 1b, and Supplementary Table 5).

In this cohort, PSA level in the plasma was also higher in prostate cancer patients ($5.4 \pm 0.2$ ng/ml) compared to control subjects ($1.0 \pm 0.1$ ng/ml) ($p < 0.0001$) and showed an AUC of 0.98 (0.96–1.00) and a diagnostic accuracy of 93% (Fig. 1c and d). At a cut-off value of 2 ng/ml, PSA sensitivity was 95% and specificity was 99%. Thus, as a single blood marker for diagnosis of prostate cancer, PSA remains as a superior marker over RNASE4. PSA amount showed a positive correlation with age, but not race in control subjects, however had no correlation with age or race in prostate cancer patients (Supplementary Table 6).

**RNASE4 enhanced the performance of PSA in prostate cancer diagnosis.** ROC curve analysis of RNASE4 plus PSA showed an excellent diagnostic performance with an AUC of 0.99 (0.98–1.00) (Fig. 1e), suggesting that combining RNASE4 with PSA may give the most accurate diagnosis of prostate cancer. Further, a positive correlation (Pearson r = 0.27, p < 0.0031) was found between PSA and RANSE4 in this cohort of prostate cancer patients (Fig. 1f). ANG is another member of the pancreatic RNase superfamily and has been shown to promote prostate cancer progression[25–31] and to be co-regulated and co-expressed with RNASE4[32,33]. We therefore also measured ANG levels in the plasma of this cohort of patients and found that plasma ANG amount was $475.1 \pm 8.6$ ng/ml in prostate cancer patients, higher than that in control subjects ($379.3 \pm 6.3$ ng/ml, $p < 0.0001$) (Supplementary Fig. 3a). These results are in agreement with previous reports that ANG improves diagnostic performance in prostate cancer screening[34]. As in case of RNASE4, ANG levels are not correlated to patient demographics (Supplementary Table 7). ROC curve analysis of ANG showed an AUC of 0.79 at the 394 ng/ml optimal cut-off value (Supplementary Fig. 3b), confirming that it is a good diagnosis marker for prostate cancer, but RNASE4 was better with a higher AUC value (0.94). Not surprisingly, we found that ANG and RNASE4 were positively correlated (Pearson $r = 0.48$, $p < 0.0001$) (Supplementary Fig. 3c) as they are known to share the same promoters and are co-expressed[33]. These data demonstrate that RNASE4 is a superior marker to ANG in prostate cancer diagnosis.

**Tissue RNASE4 level distinguishes prostate cancer from BPH.** Immunoblot (Fig. 1g) and qRT-PCR (Fig. 1h) analyses showed that RNASE4 expression is higher in prostate cancer cell lines PC-3, DU145, and LNCaP than in normal prostate epithelial cell line RWPE-1. Semi-quantitative analyses of IHC staining (Supplementary Fig. 4) of RNASE4 in a tumor microarray showed that the average RNASE4 IHC score in prostate cancer ($n = 50$), BPH ($n = 20$), and normal prostate ($n = 10$) tissues was $2.9 \pm 0.1$, $1.95 \pm 0.2$, and $1.7 \pm 0.2$, respectively (Fig. 1i). Thus, RNASE4 level in prostate cancer tissues was higher than in BPH ($p = 0.0006$) and normal prostate tissue ($p = 0.0005$). Importantly, there was no differences between normal prostate and BPH samples ($p = 0.3861$). These results suggest that RNASE4 can distinguish prostate cancer from BPH, a task PSA fails to accomplish[6].

In silico analysis of *RNASE4* mRNA utilizing Oncomine human multi-cancer datasets also revealed an up-regulation of *RNASE4* mRNA in various types of human cancers, with the

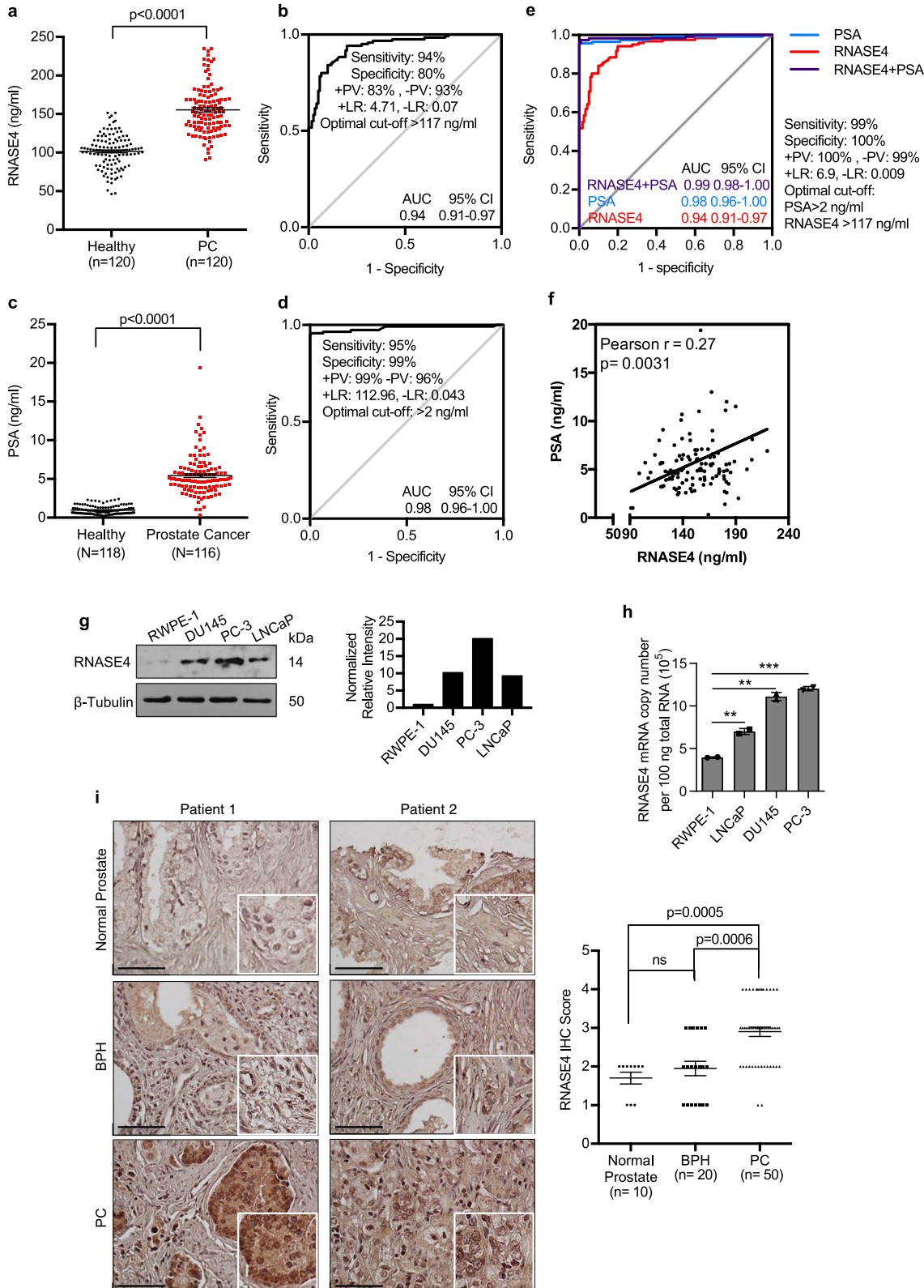

highest up-regulation observed in prostate cancer (Supplementary Fig. 5). In The Cancer Genome Atlas (TCGA) prostate dataset, *RNASE4* DNA copy number was also higher ($p = 1.94 \times 10^{-7}$) in prostate cancer tissues ($n = 171$) than in healthy tissues ($n = 61$) (Supplementary Fig. 6a). *RNASE4* DNA copy number is positively correlated with the level of *RNASE4* mRNA (Supplementary Fig. 6b). No correlation was found between *RNASE4*

transcript levels and Gleason scores of prostate cancer (Supplementary Fig. 6c). However, Kaplan-Meier analysis showed that *RNASE4* gene copy number is inversely correlated ($n = 57$, log-rank test $p = 0.03$) with recurrence-free survival (Supplementary Fig. 6d). In silico analysis also showed that prostate gland has the second highest (next to urethra) *RNASE4* mRNA level among various healthy organs (Supplementary Fig. 7a and b). These data

**Fig. 1 Up-regulation of RNASE4 in prostate cancer. a** RNASE4 protein levels in the plasma of healthy control subjects ($n = 120$) and prostate cancer (PC) patients ($n = 120$). RNASE4 amounts were determined by ELISA. Each dot represents an individual sample. Lines mark the median values and interquartile ranges. **b** Operating Characteristic Curve (ROC) analysis of RNASE4. AUC, area under the curve; +PV, positive predictive value; –PV, negative predictive value; +LR, positive likelihood ratio; -LR, negative likelihood ratio. **c** PSA levels in plasma samples of healthy control subjects ($n = 120$) and prostate cancer patients ($n = 120$). Each dot represents an individual sample. Lines mark the median values and interquartile ranges. **d** ROC analysis of PSA. **e** Combined ROC curve analysis of PSA and RNASE4 showed improved sensitivity and specificity at cut-off values 2 ng/ml and 117 ng/ml, respectively. **f** Correlation between PSA and RNASE4 in all plasma samples ($n = 240$). **g** Immunoblot analysis of RNASE4 from normal prostate epithelial cell line RWPE-1 and prostate cancer cell lines DU145, PC-3, and LNCaP. Top panel, immunoblots; bottom panel, quantification of RNASE4 protein levels by Image J using Tubulin as loading controls. **h** *RNASE4* mRNA copy numbers in 100 ng total RNA of normal prostate and cancer cell lines determined by qRT-PCR. **p ≤ 0.01 and ***p ≤ 0.001, by unpaired two-tailed Student's t test. **i** IHC analysis of RNASE4 in tissue micro array. Left panels, two sets of representative images of prostate cancer, benign prostate hyperplasia (BPH) and normal prostate tissues, scale bars = 50 μm; right panel, semi-quantitative score of RNASE4 in prostate cancer ($n = 50$), BPH ($n = 20$), and normal prostate ($n = 10$) tissues. Data were analyzed by ANOVA using Dunnett's multiple comparison test. Each dot represents score of an individual tissue sample. The horizontal lines in the plots represent the median values and the interquartile ranges.

show that *RNASE4* is highly expressed in the prostate gland and differentially enhanced in prostate cancer but not in BPH.

**RNASE4 predicts prostate biopsy outcome and is correlated with poor prognosis.** To explore the prognostic value of RNASE4, we examined the correlation between RNASE4 expression and prostate cancer aggressiveness. For this purpose, we first compared levels of RNASE4 protein in the plasma of prostate cancer patients of various clinical characteristics (Supplementary Table 8), and found that RNASE4 was positively correlated with surgical T-stage, clinical stage, biopsy grade, and surgical Gleason scores of prostate cancer patients (Fig. 2a–d), indicating that RNASE4 is associated with clinical characteristics of poor prognosis and high risk of metastasis. For example, plasma RNASE4 level in patients having pT3 tumors that extend beyond the prostate capsule was $166.8 \pm 5.6$ ng/ml ($n = 19$), higher than those having pT2 tumors that are confined in the prostate gland ($n = 75$, $149.5 \pm 2.7$ ng/ml, $p = 0.01$) (Fig. 2a). Patients at T2 clinical T-stage, representing tumors that invade one-half or less of the prostate lobe (T2a, $n = 17$, $166.6 \pm 6.7$ ng/ml, $p = 0.006$) or more than one-half of the lobe (T2b, $n = 8$, $178.5 \pm 4.2$ ng/ml, $p = 0.001$), had higher plasma RNASE4 level than those with smaller, impalpable T1 tumors (T1c, $n = 84$, $148.0 \pm 2.4$ ng/ml) (Fig. 2b). Plasma RNASE4 levels in patients with less differentiated cancer tissues, such as tumors with biopsy grade 7 ($n = 26$, $164.1 \pm 3.9$ ng/ml, $p = 0.03$) or surgical Gleason 7 ($n = 32$, $157.8 \pm 3.3$ ng/ml, $p = 0.05$), were higher than in those with well differentiated cancer tissues, such as tumors with biopsy grade 6 ($n = 89$, $149.0 \pm 2.8$ ng/ml) or surgical Gleason 6 ($n = 84$, $147.8 \pm 3.1$ ng/ml) (Fig. 2c and d). Importantly, PSA failed to correlate with aggressiveness of the tumors (Supplementary Fig. 2a–d). It is noteworthy that while plasma level of ANG was also correlated with surgical and clinical stages, it was not correlated with biopsy grade and surgical Gleason scores (Supplementary Fig. 3d–g), indicating that ANG is associated with aggressiveness of the tumor but not with their differentiation stage. These data suggest that RNASE4 is superior to PSA and to ANG as a prognosis marker of prostate cancer.

Logistic regression algorithm was used to assess whether RNASE4 can be used alone and in combination with PSA to predict biopsy outcome. Both univariate and multivariate logistic regression analyses (Table 1) showed that RNASE4 and PSA were associated with biopsy status, but only RNASE4 was associated with surgical T-stage, clinical stage, biopsy grade, and surgical Gleason. These results indicate that RNASE4 level in the plasma of prostate cancer patients is associated with advanced disease status, demonstrating the diagnostic and prognostic value of RNASE4 as an independent biomarker to predict cancer and to determine disease stage prior to biopsy.

IHC analysis of RNASE4 in prostate cancer TMA (Supplementary Fig. 4) revealed that the tissue level of RNASE4 was also positively correlated with histopathological characteristics (Supplementary Table 9) of prostate cancer patients. The average IHC score of RNASE4 was higher in prostate cancer with clinical characteristics that likely indicate a poor outcome and high risk of metastasis (Fig. 2e–i). For example, RNASE4 is higher in poorly differentiated tumors of histological grade 3 than in well-to-moderately differentiated tumors of histological grade 1 and 2 ($p = 0.0003$) (Fig. 2e), in tissues with poorly formed glands of Gleason score 4 and 5 than in predominantly well-formed glands of Gleason score 2 and 3 ($p < 0.0001$) (Fig. 2f), in advanced invasive tumors of T3 and T4 stage than in less invasive tumors of T1 and T2 stage ($p = 0.005$) (Fig. 2g), and in those with distant or lymph node metastasis than in those with no metastasis ($p = 0.0041$ and $0.0001$, respectively) (Fig. 2h and i). These results strongly suggest that RNASE4 can predict prostate biopsy outcome, and is correlated with poor prognosis and patient survival, and may thus serve as a prognosis marker.

**RNASE4 induces prostate cancer cell proliferation by activating PI3K-AKT-mTOR pathway.** To investigate the functional role of RNASE4 in prostate cancer, we examined the effects of exogenous RNASE4 on prostate cancer cells, and found that RNASE4 stimulates proliferation of DU145 (Fig. 3a) and PC-3 cells (Supplementary Fig. 8a). Cell cycle analysis of DU145 cells showed that a 3-day treatment with RNASE4 decreased G0/G1 and increased G2/S/M populations (Fig. 3b). To investigate the mechanism of RNASE4-induced cell proliferation, we examined the effects of exogenous RNASE4 on the PI3K-AKT-mTOR pathway that has been known to play a key role in prostate cancer[35], and found that AKT and S6 were rapidly and continuously activated by RNASE4 in DU145 (Fig. 3c) and PC-3 cells (Supplementary Fig. 8b), which could be inhibited by AKT inhibitor MK-2206, PI3K inhibitors LY294002 and Wortmannin (Fig. 3d). It is notable that Rapamycin, an mTOR inhibitor, inhibited S6 phosphorylation but not AKT phosphorylation. Consistently, RNASE4-induced cell proliferation was abolished by all these inhibitors (Fig. 3e and Supplementary Fig. 8c). These data suggest that RNASE4 stimulates prostate cancer cell proliferation likely by activating the PI3K-AKT-mTOR-S6K signaling pathway.

**RNASE4 activates AXL to stimulate prostate cancer cell proliferation.** Since RNASE4 is a secreted protein, the above findings suggest a receptor-mediated function of RNASE4. We performed human phospho-receptor tyrosine kinases (RTK) antibody array screening and found that RNASE4 treatment induced phosphorylation of AXL in DU145 (Fig. 4a) and PC-3

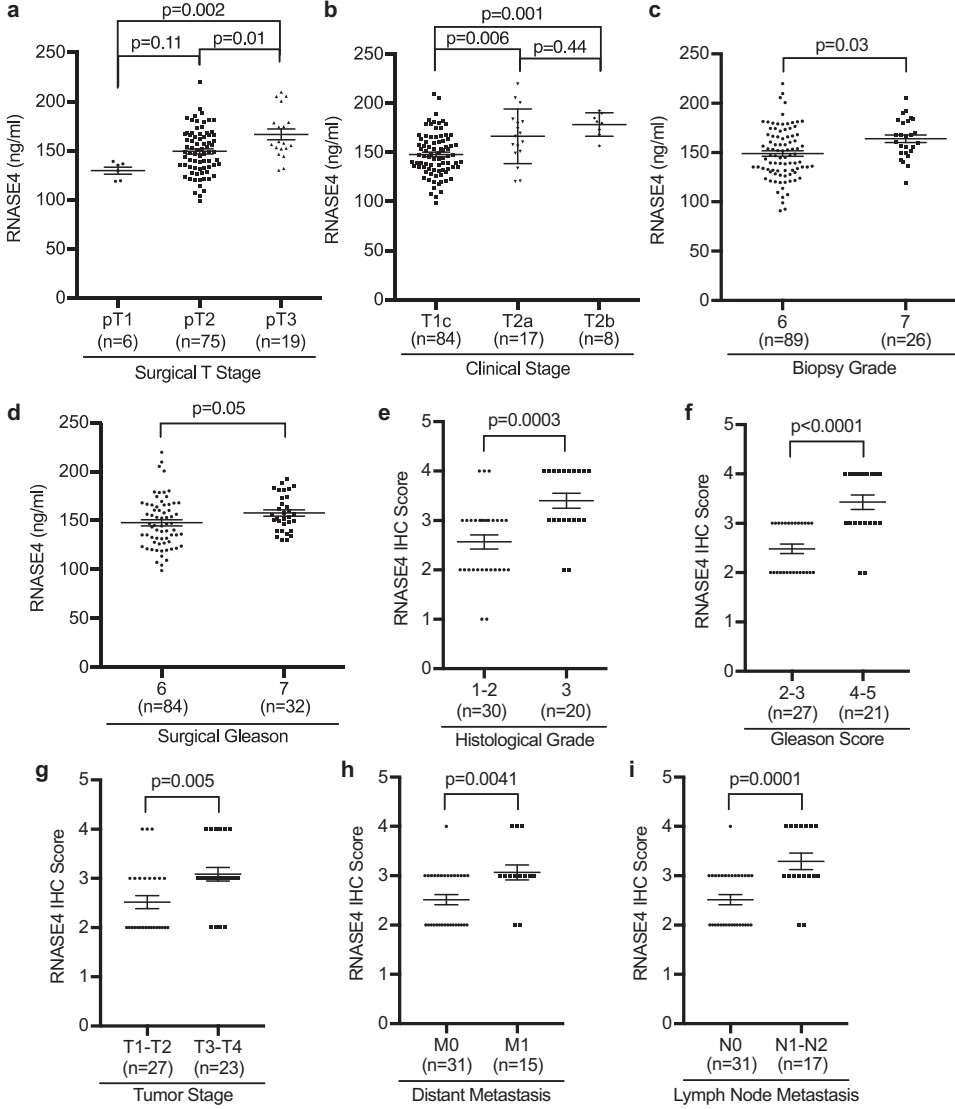

**Fig. 2 RNASE4 protein levels in the plasma tissues of prostate cancer patients are correlated with poor prognosis and high risk of metastasis.**
**a–d** Correlation of plasma RNASE4 protein levels with tumor surgical T-stage (**a**), clinical stage (**b**), biopsy grade (**c**), and surgical Gleason (**d**). RNASE4 levels in the plasma were determined by ELISA. **e–i** Correlation of tissue RNASE4 levels with tumor histological grade (**e**), Gleason score (**f**), tumor stage (**g**), distant metastasis (**h**), and lymph node metastasis (**i**). Tissue RNASE4 levels were determined by semi-quantitative IHC. Each dot represents an individual sample. Lines represent the median values and the interquartile ranges. Statistical analyses were done by one-way ANOVA (**a**, **b**) and two-tailed Student's *t* test (**c–i**).

**Table 1 Univariate and multivariate regression analysis of RNASE4 and PSA predictive of biopsy outcome.**

| Outcome variable | Predictor variable | Univariate | | Multivariate | |
|---|---|---|---|---|---|
| | | OR (95% CI) | *P*-value | OR (95% CI) | *P*-value |
| Biopsy status: Prostate cancer (*n* = 120) vs. healthy (*n* = 120) | PSA | 15.97 (6.44–39.60) | <0.0001 | 9.34 (3.90–22.38) | <0.0001 |
| | RNASE4 | 1.10 (1.08–1.13) | <0.0001 | 1.07 (1.03–1.12) | <0.0001 |
| Surgical T-stage: pT3 (*n* = 19) vs. pT2 (*n* = 75) | PSA | 1.10 (0.95–1.29) | 0.19 | 1.08 (0.92–1.27) | 0.32 |
| | RNASE4 | 1.03 (1.01–1.05) | 0.001 | 1.03 (1.01–1.05) | 0.0007 |
| Clinical stage: T2 (RNASE4, *n* = 25; PSA, *n* = 26) vs. T1 (*n* = 84) | PSA | 1.00 (0.86–1.17) | 0.97 | 0.95 (0.80–1.13) | 0.50 |
| | RNASE4 | 1.04 (1.02–1.06) | 0.0002 | 1.04 (1.02–1.06) | <0.0001 |
| Biopsy grade: 7 (*n* = 26) vs. 6 (RNASE4, *n* = 89; PSA *n* = 87) | PSA | 1.06 (0.90–1.24) | 0.47 | 1.01 (0.85–1.20) | 0.92 |
| | RNASE4 | 1.03 (1.00–1.04) | 0.01 | 1.02 (1.00–1.04) | 0.01 |
| Surgical gleason 7 (*n* = 32) vs. 6 (RNASE4, *n* = 84; PSA *n* = 66) | PSA | 1.04 (0.86–1.27) | 0.64 | 1.02 (0.84–1.26) | 0.73 |
| | RNASE4 | 1.02 (0.99–1.04) | 0.05 | 1.02 (0.99–1.04) | 0.07 |

*OR* odds ratio, *CI* confidence interval. Equal *n* reported for PSA and RNASE4 if not stratified.

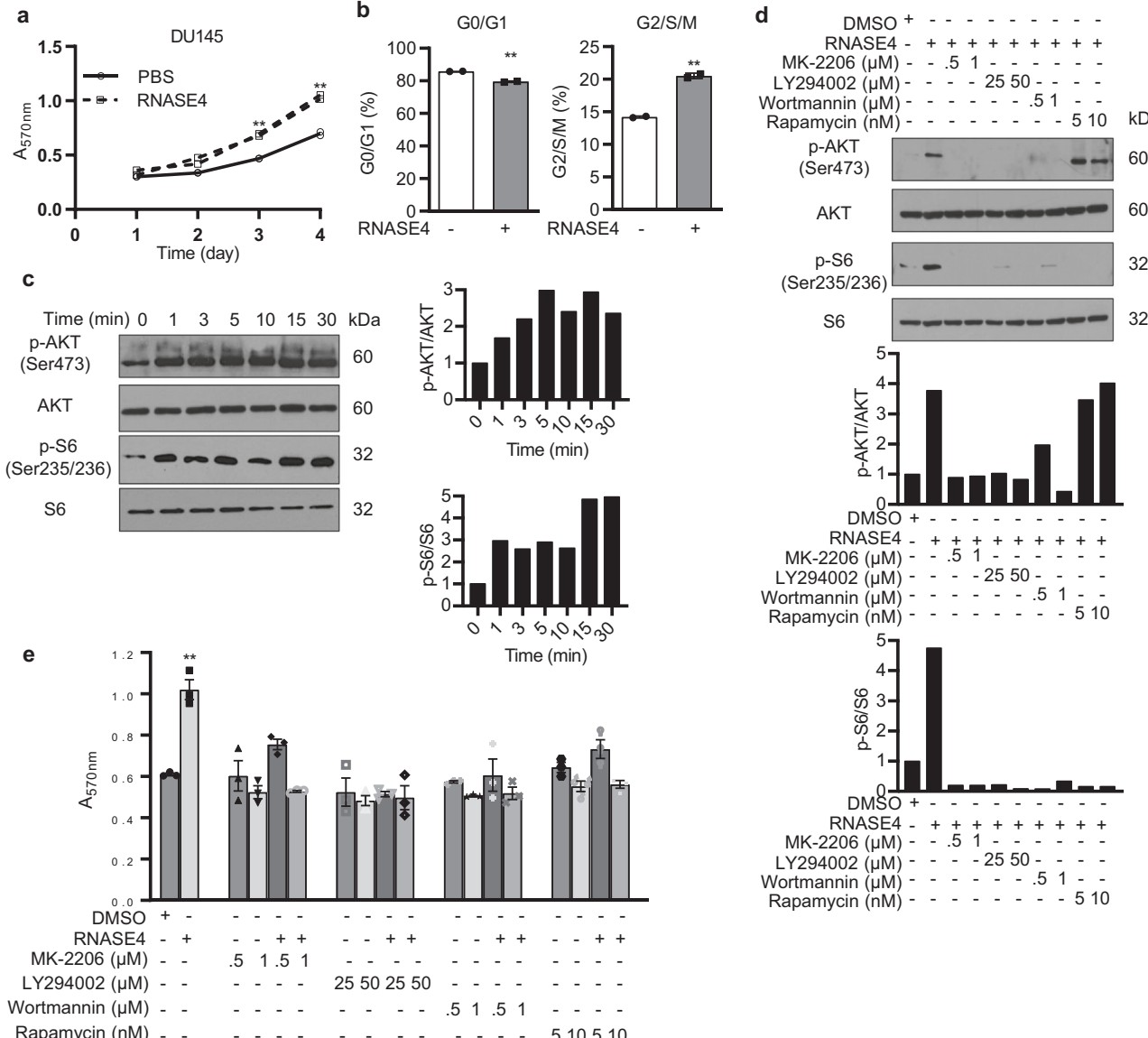

**Fig. 3 RNASE4 induces prostate cancer cell proliferation and phosphorylation of AKT and S6. a, b** Exogenous RNASE4 (1 µg/ml) stimulates cell proliferation (**a**) and cell cycle progression (**b**) of DU145 cells cultured in the presence of 2% FBS. Cell numbers were determined by MTT assay. Cell cycle status was determined by flow cytometry after 24 h incubation with RNASE4. **c** Time course of AKT and S6 phosphorylation of serum-starved DU145 cells by RNASE4 (1 µg/ml). Left panels, immunoblots; right panels, quantification of p-AKT and p-S6 normalized to total AKT and S6 by Image J analysis. **d** Effect of AKT inhibitor MK-2206, PI3K inhibitors Wortmannin and LY294002, and mTOR inhibitor Rapamycin on RNASE4-induced phosphorylation of AKT and S6. Cells were pre-incubated with the inhibitors at the indicated concentrations for 1 h prior to be stimulated by 1 µg/ml of RNASE4 for 5 min. Top panels, immunoblots; bottom panels, Image J quantification of p-AKT and p-S6 normalized to total AKT and S6. **e** Effect of AKT inhibitor MK-2206, PI3K inhibitors Wortmannin and LY294002, and mTOR inhibitor Rapamycin on RNASE4-induced cell proliferation. DU145 cells were serum-starved overnight, incubated with the inhibitors for 1 h, and then treated with RNASE4 (1 µg/ml) for three days. Cell numbers were determined by MTT assay. Data shown are from a representative experiments in triplicates of 3 independent repeats. Error bars indicate SEM. **$p \leq 0.01$, by unpaired two-tailed Student's $t$ test.

(Supplementary Fig. 8d) cells. AXL is a mediator of cell growth and survival[36], and is up-regulated in several cancers[37–39], including prostate cancer[40–42]. RNASE4-induced AXL phosphorylation was confirmed by immunoblot analyses in DU145 (Fig. 4b) and PC-3 cells (Supplementary Fig. 8e).

Next, we examined the effect of R428, a selective small molecule inhibitor of AXL kinase, on RNASE4-induced DU145 and PC-3 cell proliferation, and found that it inhibited RNASE4-induced cell proliferation (Fig. 4c and Supplementary Fig. 8f) and phosphorylation of AKT and S6 (Fig. 4d). These results suggest a relationship between RNASE4 and AXL in regulating AKT and S6 phosphorylation in prostate cancer cells.

**RNASE4 knockdown suppresses prostate cancer cell proliferation and tumor growth.** We next examined the cell-autonomous function of RNASE4 in prostate cancer proliferation by knocking down *RNASE4* in PC-3 (Fig. 5a and b), DU145 (Supplementary Fig. 9a and b), and LNCaP (Supplementary Fig. 10a and b) cells using two *RNASE4*-specific shRNAs (shRNASE4-1 and shRNASE4-2) with a non-targeting shRNA control (shControl). Knockdown of *RNASE4* decreased cell proliferation (Fig. 5c, Supplementary Figs. 9c and 10c) and reduced both the number and the size of prostate cancer cell colonies in soft agar (Fig. 5d, Supplementary Figs. 9d and 10d). Exogenous RNASE4 was able to rescue the effect of *RNASE4* knockdown in

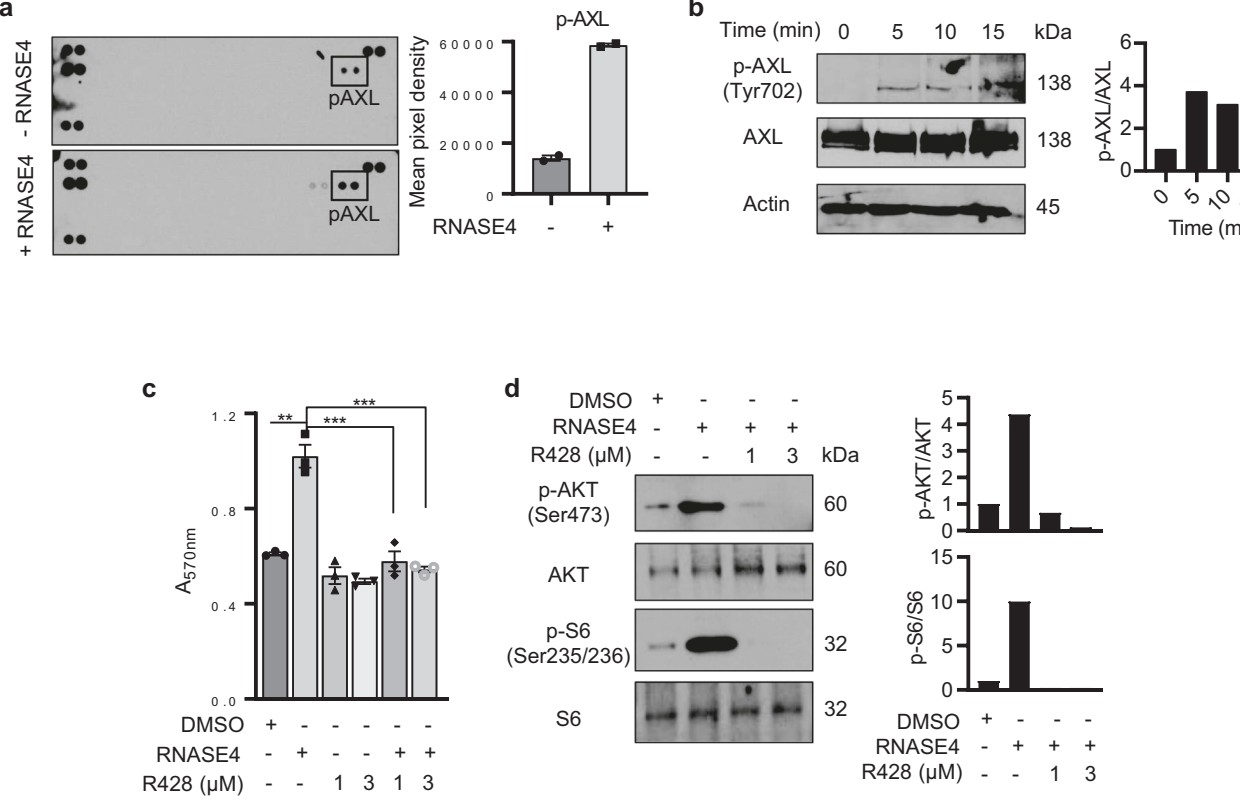

**Fig. 4 RNASE4 induces AXL phosphorylation. a** Human phospho-receptor tyrosine kinase (RTK) antibody array analysis. Serum-starved DU145 cells were treated with or without 1 μg/ml of RNASE4 for 5 minutes. A total of 1.5 mg cell lysate protein was blotted on each array membrane. The right panel shows the mean pixel intensity of p-AXL. **b** Immunoblot analysis of RNASE4-stimulated AXL phosphorylation. DU145 cells were serum-starved overnight and treated with 1 μg/ml of RNASE4 for the indicated time. Cell lysates were analyzed by antibodies against total AXL and phospho-AXL, with β-actin as the loading control. Left panels, immunoblots; right panel, Image J quantification of p-AXL normalized to total AXL. **c** Effect of AXL inhibitor R428 on RNASE4-induced cell proliferation. Serum-starved DU145 cells were incubated with 1 or 3 μM R428 for 3 h and then stimulated by 1 μg/ml of RNASE4 for 3 days. Cell numbers were determined by MTT assay. Data shown are means ± SEM of a representative experiment (in triplicates) of 3 independent repeats. **p ≤ 0.01 and ***p ≤ 0.001, by unpaired two-tailed Student's t test. **d** Effect of AXL inhibitor R428 on RNASE4-induced AKT and P6 phosphorylation. Serum-starved DU145 cells were incubated with 1 or 3 μM R428 for 3 h and then stimulated by 1 μg/ml of RNASE4 for 5 min. Cell lysates were analyzed for AKT and P6 phosphorylation by immunoblotting. Top panels, immunoblots; bottom panels, quantification of phospho-AKT and phospho-S6 by Image J analyses normalized to total AKT and P6, respectively.

both cell proliferation (Fig. 5c, Supplementary Figs. 9c and 10c) and colony formation assays in soft agar (Fig. 5d, Supplementary Figs. 9d and 10d). Next, we examined if ribonucleolytic activity of RNASE4 was essential to induce cell proliferation. For this purpose, RNASE4 protein was treated with 2 mM of diethyl pyrocarbonate (DEPC) in PBS for 10 min, a method known to inactivate RNase by modifying His and Lys residues in the catalytic site[43]. DEPC-treated RNASE4 (Fig. 5e) had no enzymatic activity using yeast tRNA as the substrate (Fig. 5f), and was not able to rescue the effect of *RNASE4* knockdown in cell proliferation (Fig. 5g). These results indicate that the ribonucleolytic activity is essential for RNASE4 to mediate cell proliferation.

The effect of *RNASE4* knockdown on tumor growth was examined in a xenograft animal model with PC-3 cells. Compared to mice injected with shControl-transfected cells, tumor growth was slower in mice inoculated with shRNASE4-transfected cells as reflected by a 75 ± 6% decrease in tumor volume (Fig. 5h, p = 0.011) and 88 ± 3% decrease in tumor weight on day 23 when animals were sacrificed (Fig. 5i, p = 0.03). Consistent with AXL being a mediator for RNASE4, we observed a decreased level of phosphorylated AXL in *RNASE4* knockdown tumors while the level of total AXL was not changed (Fig. 5j). A reduction of RNASE4 protein was confirmed in tumors derived from shRNASE4-transfected cells compared to those derived from

shControl-transfected cells (Fig. 5k). The percentage of Ki-67 and CD31 positive cells were also decreased in *RNASE4* knockdown tumors (Fig. 5k), indicating a decrease in cell proliferation and angiogenesis upon *RNASE4* knockdown.

**RNASE4 mAb inhibits prostate cancer cell proliferation and RNASE4-induced angiogenesis.** To evaluate the therapeutic value of targeting RNASE4 in prostate cancer therapy, we examined the effect of human RNASE4-specific mAb on prostate cancer cell proliferation. As shown in Fig. 6a, RNASE4 mAb inhibited proliferation of DU145, PC-3, and LNCaP cells in a dose-dependent manner. Consistently, it decreased the population of cells in G2/S/M phase and correspondingly increased G0/G1 population (Fig. 6b). RNASE4 is known to be angiogenic[23]. We therefore examined the effect of RNASE4 mAb on RNASE4-induced angiogenesis by in vitro endothelial cell tube formation assay, and found that it inhibited RNASE4-induced endothelial cell tube formation (Fig. 6c) in a dose-dependent manner (Supplementary Fig. 11), but had no effect to endothelial cell tube formation induced by basic fibroblast growth factor (bFGF), an unrelated angiogenic factor (Supplementary Fig. 12). It is currently unclear if RNASE4 induces cell proliferation and angiogenesis by the same mechanism, but these results demonstrate the

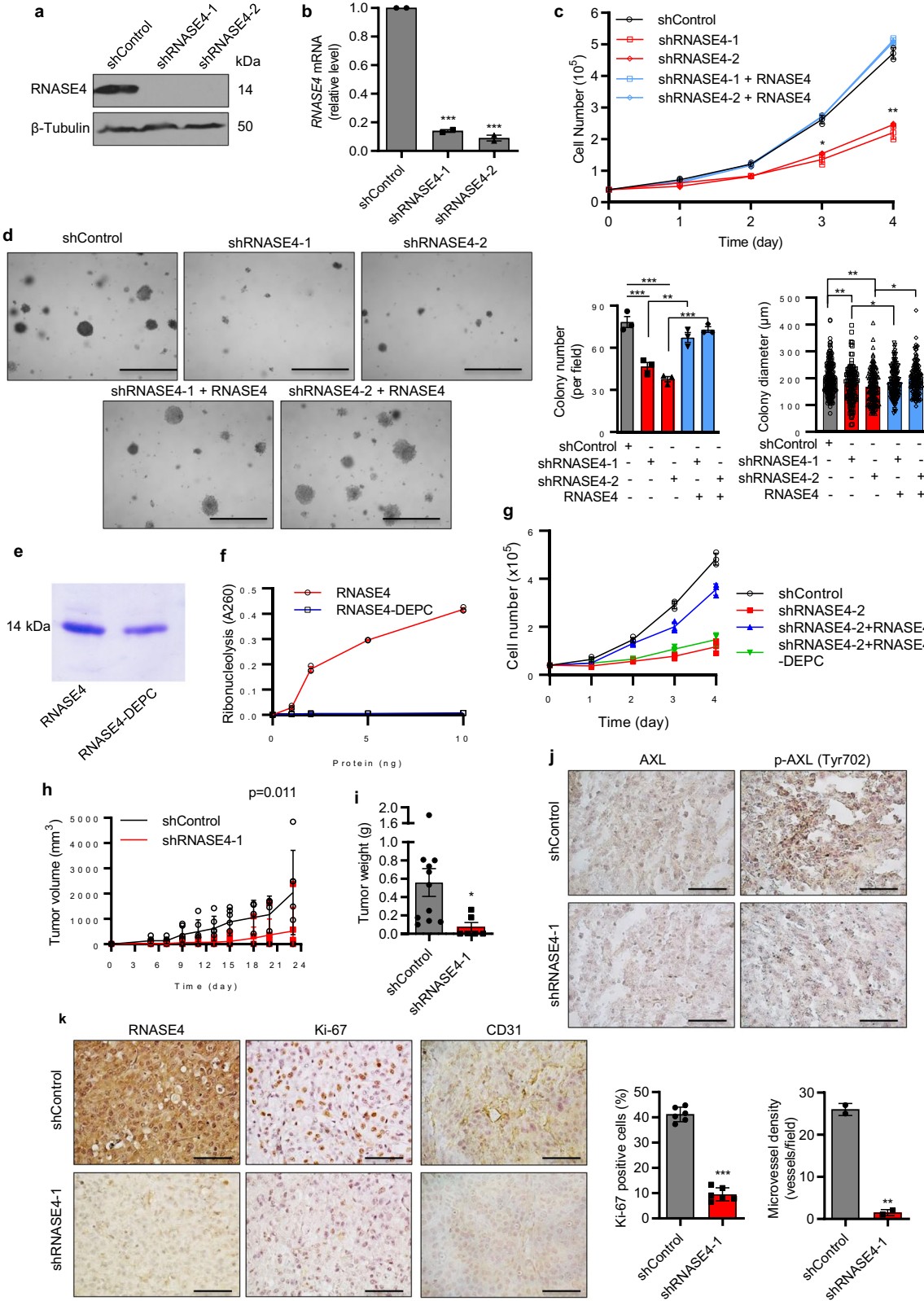

effectiveness and specificity of RNASE4 mAb in inhibiting the activity of RNASE4 in these cellular events.

**RNASE4 mAb suppresses prostate tumor growth in vivo.** The anti-tumor activity of RNASE4 mAb was first examined in a prophylactic setting in a xenograft animal model in which PC-3 cells were inoculated into athymic mice and treatment was started

the next day. Antibody was administrated by i.p. injection at 10 mg/kg once every 3 days for 63 days. This treatment regimen resulted in an $80 \pm 5\%$ inhibition tumor volume (Fig. 7a, $p = 0.0014$) a $90 \pm 3\%$ inhibition in tumor weight (Fig. 7b, $p = 0.05$). Body weight of the animals was not changed (Fig. 7c) upon mAb treatment, suggesting no or low toxicity by inhibiting RNASE4. IHC analysis of Ki-67 and CD31 showed that cancer

**Fig. 5 _RNASE4_ knockdown reduces prostate cancer cell proliferation in vitro, in soft agar, and in athymic mice. a** Immunoblot analysis of RNASE4 in PC-3 cells stably transfected with shControl, shRNASE4-1, and shRNASE4-2. **b** qRT-PCR analysis of _RNASE4_ mRNA normalized to _GAPDH_ in PC-3 cells stably transfected with shControl, shRNASE4-1, and shRNASE4-2. **c** Effect of _RNASE4_ knockdown on cell proliferation. Cell numbers were counted by a Coulter counter. Exogenouse RNASE4, when presented, was at the concentration of 1 μg/ml. Data shown are a representative experiment (in triplicates) of 3 independent repeats. **d** Effects of _RNASE4_ knockdown on colonies formation and growth of PC-3 cells in soft agar. The same numbers of control and _RNASE4_ knockdown cells were seeded in soft agar, and grown in the absence or presence of 1 μg/ml of RNASE4 for 14 days. Colonies were counted and size measured on a Nikon Eclipse _ti_ microscope. Left panels, representative images of colonies; scale bars =100 μm. Right panels, quantification of colony numbers and sizes. **e** SDS-PAGE of RNASE4 and RNASE4-DEPC. **f** Ribonucleolytic activity of RNASE4 and RNASE4-DEPC measured by the tRNA cleavage method. **g** Effect of RNASE4 and RNASE4-DEPC (1 μg/ml) on proliferation of _RNASE4_ knockdown PC-3 cells. Cell numbers were determined by MTT assay. Data shown are a representative experiment (in triplicates) of 3 independent repeats. **h** Growth curve of PC-3 control and _RNASE4_ knockdown cells in athymic mice. Same number of shControl- and shRNASE4-1-tranfected PC3 cells (1 × 10⁶ per mouse) were inoculated s.c. on the right lower back of nude mice (n = 6 per group). Tumor volume was measured two times per week. **i** Tumor weight derived from control and _RNASE4_ knockdown PC-3 cells. On day 24, mice were sacrificed, and their tumors were dissected and weighed. **j** IHC analyses of total AXL and phosphoylated AXL (p-AXL) in tumor sections derived from control and _RNASE4_ knckdown PC-3 cell. **k** IHC analyses of RNASE4, Ki-67, and CD31 in tumor sections derived from control and _RNASE4_ knckdown PC-3 cell. Quantification of Ki-67 positive cells and CD31 positive neovessels were from 5 randomly selected microscopic fields. Scale bars =100 μm. Data shown are means ± SEM. Scale bars =100 μm. *$p \leq 0.05$, **$p \leq 0.01$ and ***$p \leq 0.001$, by two-tailed Student's _t_ test.

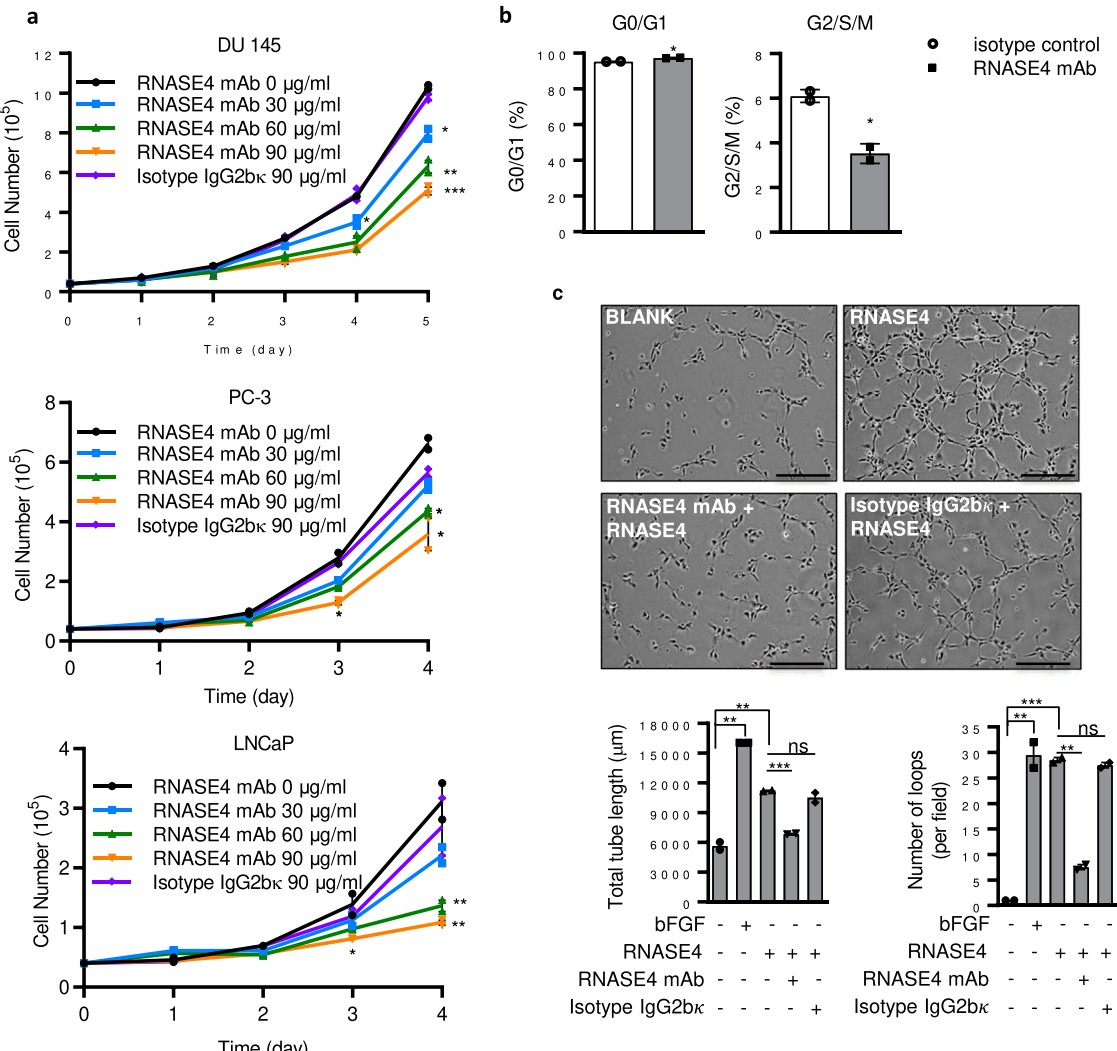

**Fig. 6 RNASE4 mAb inhibits prostate cancer cell proliferation and RNASE4-induced angiogenesis. a** Effect of RNASE4 mAb on cell proliferation. DU145, PC-3, and LNCaP cells were cultured in the presence of 2% FBS. RNASE4 mAb or isotype control IgG2bκ were added at the concentrations indicated. Cell numbers were determined by a Coulter counter. **b** Effect of RNASE4 mAb on cell cycle distribution of DU145 cells. Cells were cultured in the 2% FBS and RNASE4 mAb (30 μg/ml) or isotype control IgG2bκ (30 μg/ml) for 3 days. Cell cycle status was determined by flow cytometry. **c** Effect of RNASE4 mAb on endothelail cell tube formation. HUVEC were cultured on Matrigel-coated wells and incubated in the absence or presence of 1 μg/m of RNASE4 with 30 μg/ml of RNASE4 mAb or isotype control IgG2bκ for 4 h. Top panels, images of endothelial cell tubular structure of a representative experiment in duplicate of 3 independent repeats. Scale bar = 50 μm. Bottom panels, tube length and number of loops counted from 5 randomly selected areas. Data shown are means ± SEM. *$p \leq 0.05$, **$p \leq 0.01$ and ***$p \leq 0.001$, by two-tailed Student's _t_ test.

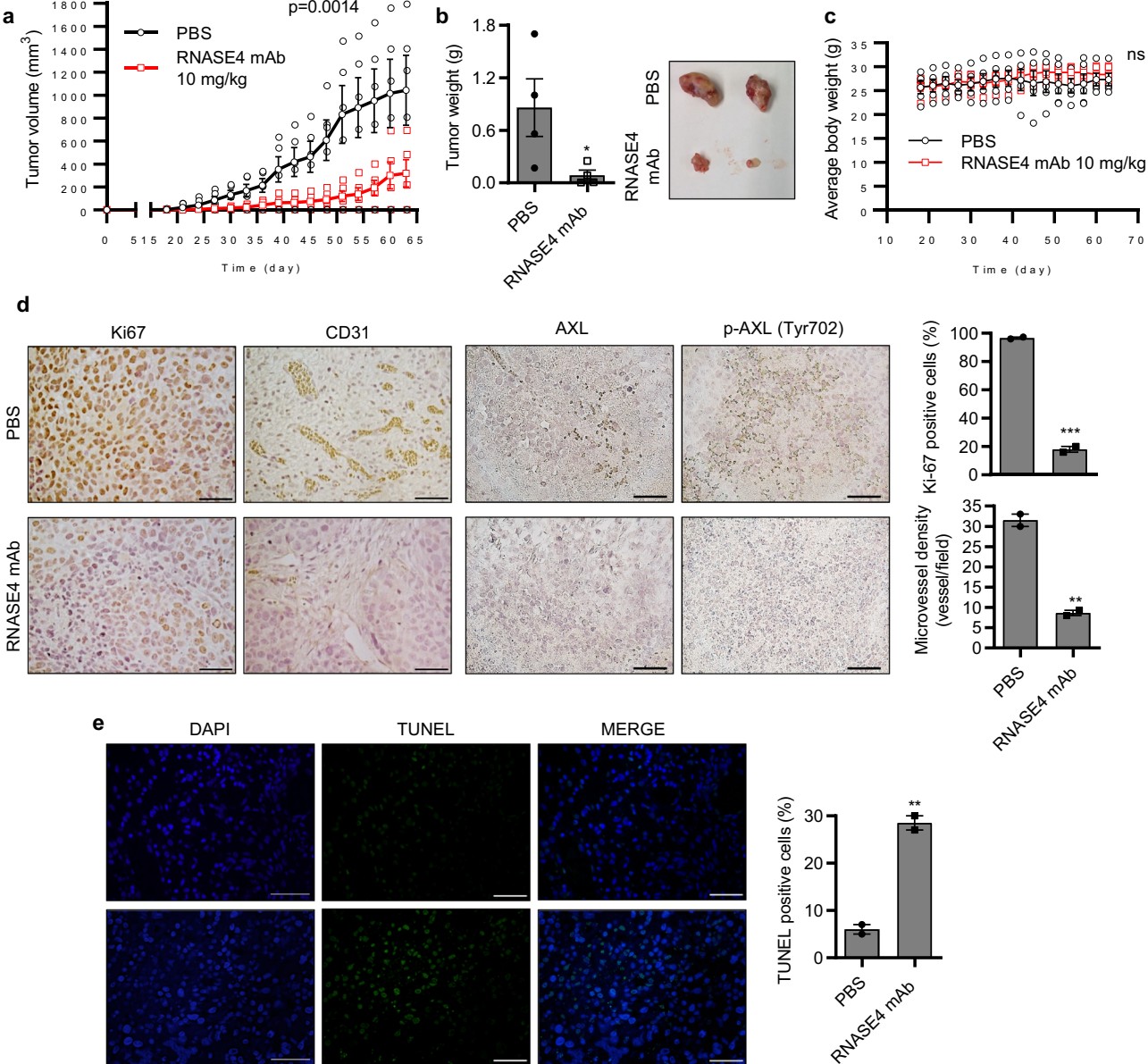

**Fig. 7 RNASE4 mAb inhibits the establishment of xenograft human prostate cell tumors in athymic mice.** PC-3 cells (1 × 10⁶ per mouse) were inoculated s.c. on the back of male athymic mice. One day post-inoculation, mice were treated by i.p. injection with PBS (*n* = 6) or RNASE4 mAb (10 mg/kg, *n* = 6) once every 3 days. **a** Tumor volume measured once every 3 days. **b** Animals were sacrificed on day 64, tumors were dissected and weighed. Left panel, tumor weight; right panel, representative images of tumors from PBS- and RANSE4 mAb-treated groups. **c** Body weight of PBS- and RNASE4 mAb-treated animals. **d** IHC analyses of Ki-67, CD31, total AXL, and phophoylated AXL (p-AXL) in tumor sections from PBS- and RNASE4 mAb-treated animals. Left panels, representative images; right panels, quantification of Ki-67 positive cells and CD31 positive neovessels from 5 randomly selected microscopic fields. **e** TUNEL staining of apoptotic cells. Nuclei were stained by DAPI. Right panels, representative images; right panel, percentage of TUNEL positive cells counted in five randomly selected areas. Data shown are means ± SEM. Scale bars = 100 μm. **p ≤ 0.01, ***p ≤ 0.001, and n.s. = not significant, by two-tailed Student's *t* test.

cell proliferation and tumor angiogenesis were reduced (Fig. 7d), while TUNEL staining showed that apoptosis was enhanced (Fig. 7e) upon mAb treatment. A reduction in phospho-AXL was also observed in RNASE4 mAb-treated tumors (Fig. 7d). These results indicate that RNASE4 mAb inhibited the initiation and growth of PC-3 xenograft tumors in athymic mice, accompanied by a reduction in tumor cell proliferation, tumor angiogenesis, and an increase in tumor cell apoptosis, as well as the involvement of AXL in these processes.

Next, we evaluated the effectiveness of RNASE4 mAb against established tumors. PC-3 cells were inoculated into athymic mice

and waited for 21 days until xenograft tumors reached a size of approximately 100 mm³. The tumor-bearing animals were separated into 3 groups according to tumor sizes so that each group had animals with matched tumor sizes, and treated with PBS (*n* = 12) or with RNASE4 mAb at 10 mg/kg (*n* = 6) or 30 mg/kg (*n* = 6), respectively, by i.p. injection once every 3 days for 39 days. RNASE4 mAb dose-dependently inhibited growth of established PC-3 xenograft tumors in athymic mice. Treatment with 30 mg/kg resulted in a reduction of 72.7 ± 6% in tumor volume (Fig. 8a) and 70.0 ± 3% in tumor weight (Fig. 8b). Again, IHC analyses showed that RNASE4 mAb-treated tumors displayed a reduced staining of

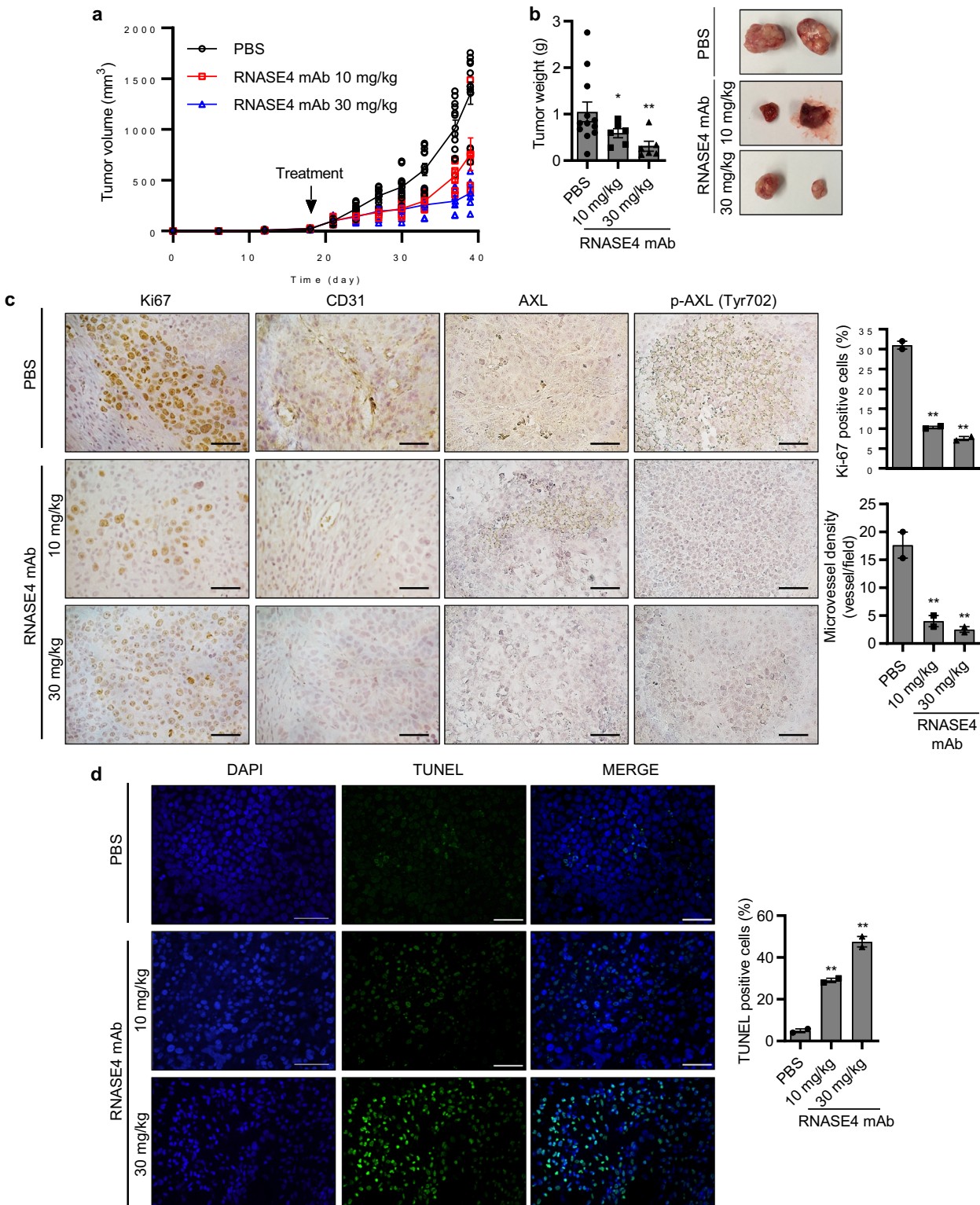

**Fig. 8 RNASE4 mAb inhibits the growth of established xenograft human prostate cell tumors in athymic mice.** PC-3 cells ($1 \times 10^6$) were inoculated s.c. on the back of male athymic mice. After tumors reached ~100 mm³ in volume (day 21), mice were grouped and treated by i.p. injection of PBS ($n = 12$) or RNASE4 mAb (10 mg/kg or 30 mg/kg, $n = 6$ per group) three times per week. **a** Tumor volume measured every 3 days. **b** Tumor weight. Mice were sacrificed on day 39, tumors were dissected and weighed. Left panel, average tumor weight; right panel, representative tumor images from the three groups. **c** IHC analyses for ki-67, CD31, total AXL, and phosphorylated AXL (p-AXL). Left panels, representative images; right panels, quantification of Ki-67 positive cells and CD31 positive neovessels from 5 randomly selected microscopic fields. **d** TUNEL staining of apoptotic cells. Nuclei were stained by DAPI. Right panels, representative images; right panel, percentage of TUNEL positive cells in five randomly selected areas. Data shown are means ± SEM. Scale bars = 100 μm. *$p \leq 0.05$ and **$p \leq 0.01$, by two-tailed Student's $t$ test.

Ki67, CD31, and phosphorylated AXL (Fig. 8c), and an increased TUNEL staining (Fig. 8d). Taken together, these results provide in vivo evidence that RNASE4 is a therapeutic target for the treatment of prostate cancers.

## Discussion

The main finding of this study is that RNASE4, a ribonuclease previously unknown to have any involvement in prostate cancer, was found to play a role in prognosis, diagnosis, and treatment of prostate cancer. It is notable that the plasma level of RNASE4 protein has the potential to serve as an accurate blood biomarker for prostate cancer to predict disease aggressiveness. It is also notable that RNASE mAb inhibits both the establishment of xenograft tumors and growth of established tumors in mice.

Prognostic and predictive biomarkers for prostate cancer are an important unmet medical need for deciding the best treatment option and for tracking disease progression. PSA has been controversial for this purpose due to its limitations such as poor specificity, lack of differentiation between prostate cancer and BPH, and inability to separate aggressive and indolent tumors[44]. Our study shows that RNASE4 is a new serum and tissue biomarker for prostate cancer. While PSA is superior to RNASE4 when used as an individual serum marker for prostate cancer, RNASE4 has the potential to distinguish cancer from benign growth. While there is an elevated RNASE4 level in prostate cancer tissue samples compared to BPH tissue samples ($p = 0.0006$), no differences were observed in RNASE4 levels between healthy controls and BPH samples ($p = 0.3861$), indicating the superior specificity of RNASE4 in prostate cancer diagnosis. Since PSA and PSA derivatives, such as PSA velocity, PSA density and free PSA fraction have all proven to be disappointing in distinguishing between BPH and prostate cancer[44], RNASE4 represents a valuable alternative to these tests.

Our results also demonstrate a correlation between the plasma RNASE4 level and tumor grade, stage, metastasis and prognosis. Thus, plasma RNASE4 level can provide valuable information on cancer aggressiveness and can predict biopsy outcome, and may stratify patients for the need of biopsy. Prostate biopsy is an invasive and painful procedure that can lead to complications; therefore minimizing number of biopsies by alternative non-invasive testing methods is valuable. For instance, RNASE4 can potentially serve as a complementary test in pre-biopsy or pre-operative staging to make an educated decision about the need for needle-biopsy and about what type of treatment is the best option. The median RNASE4 amount is higher in men with higher Gleason score, biopsy grade, clinical stage, surgical T-stage and with distant and lymph node metastasis. These results suggest that RNASE4 could be a useful biomarker for predicting the outcome of both localized and lethal, metastatic prostate cancer cases. Indeed, univariate and multivariate regression analyses confirmed the significance of RNASE4 as a predictive biomarker of cancer outcome either used alone or together with PSA.

Besides the prognostic and diagnostic value of RNASE4 in prostate cancer, this study also demonstrated a functional role of RNASE4 in promoting prostate cancer progression by stimulating cancer cell proliferation and probably also by promoting angiogenesis. We have reported previously that RNASE4 was able to stimulate angiogenesis in an in vitro assay[23]. Now we show that RNASE4 mAbs inhibited RNASE4-induced angiogenesis in vitro (Fig. 6c and Supplementary Fig. 11) and decreased neovessel density in the tumor tissue (Figs. 7d and 8c). These data suggest a role of RNASE4 in angiogenesis, a function similar to that of ANG[31].

We also demonstrated that RNASE4 activates AKT and S6 in prostate cancer, suggesting that the PI3K-AKT-mTOR signaling pathway, known to play an important role in several cancers[45,46],

including prostate cancer[35], was involved in RNASE4-stimulated cell proliferation. Our data also showed that the AXL receptor kinase is activated by RNASE4. At this time, we do not know whether or not RNASE4 physically interacts with AXL to induce its auto-phosphorylation. However, it is clear that AXL is required for RNASE4 to exert its function in inducing cell proliferation as its specific inhibitor R428 abolished RNASE4-induced AKT phosphorylation and cell proliferation. AXL is reported to be overexpressed in multiple cancers[37–39], including prostate cancer[40–42]. Several studies have shown that AXL overexpression may result in resistance to both targeted therapies and conventional chemotherapy in cancer[46,47]. Indeed, AXL has been shown to be up-regulated in metformin-resistant prostate cancer cells[42]. AXL overexpression in cancer is also correlated with poor prognosis and increased metastasis[47], and AXL-mediated activation of PI3K-AKT-mTOR pathway has been linked to tumor progression and to increased proliferation of prostate cancer cells[40,41]. Consistent with a role of AXL in RNASE4-induced prostate cancer growth, we found that RNASE4 knockdown or treatment with RNASE4 mAb in xenograft animal models resulted in a reduction of AXL phosphorylation without affecting the total AXL levels in the tumor tissues.

Another notable finding of this study is that RNASE4 may serve as a therapeutic target for drug development in the treatment of prostate cancer. A RNASE4-specific mAb has been shown to inhibit prostate cancer cell growth in vitro, in soft agar, and in athymic mice. It is effective in both prophylactic and therapeutic settings in inhibiting xenograft growth of human prostate cancer cells in athymic mice, both accompanied by a substantial reduction in cancer cell proliferation and tumor angiogenesis, and an increase in cancer cell apoptosis.

To our knowledge this is the first study to explore the function and mechanism of RNASE4 in prostate cancer. RNASE4 represents as a new diagnostic biomarker for prostate cancer and can distinguish cancer from BPH. Most importantly, RNASE4 is associated with diseases of higher Gleason, higher biopsy grade, higher tumor stage, and higher risk of metastasis, and may thus serve as a prognosis marker. It is remarkable that a single molecule can serve both as a biomarker of prognosis and diagnosis, and as a therapeutic target for drug development. RNASE4-specific mAb may thus have an impact in prostate cancer diagnosis and treatment.

## Methods

**Human plasma samples.** A total of 240 plasma samples (120 healthy, 120 prostate adenocarcinoma) that were collected at Johns Hopkins University was obtained from Prostate Cancer Biorepository Network (PCBN), in compliance with institutional guidelines, as approved by the Institutional Review Board of Tufts Medical Center/Tufts Medical School and Johns Hopkins University School of Medicine. Informed consent was obtained for all samples.

**Tumor xenograft experiments.** All animal work was done in accordance with protocols approved by the Institutional Animal Care and Use Committee of Tufts Medical Center/Tufts Medical School. Eight-week-old male athymic (Taconic Biosciences) were injected s.c. in the right flank with $1 \times 10^6$ viable PC-3 cells in 100 μl HBSS. For the prophylactic treatment, one day after PC-3 cell inoculation, mice were administered with RNASE4 mAb (10 mg/kg) or PBS once every 3 days by i.p. injection until the end of the experiment. For the therapeutic treatment, mice were separated into treatment groups once tumors reached ~100 mm³ and treated with i.p. injection of PBS or RNASE4 mAb at 10 or 30 mg/kg once every 3 days until the end of the experiment.

**Recombinant human RNASE4 protein and RNASE4 antibodies.** Recombinant human RNASE4 protein was generated using a pET11 expression system in E.coli and purified by SP-Sepharose and reversed-phase HPLC[23]. The concentration of RNASE4 in all experiments was 1 μg/ml unless indicated otherwise. RNASE4 polyclonal antibodies (pAb) were generated using RNASE4 recombinant protein as the antigen, and affinity purified using 2.5 mg pure recombinant RNASE4 protein coupled to HiTrap NHS-activated HP column (GE Healthcare) according to manufacturer's instructions. Coupling efficiency was determined as 90% by

Bradford Assay. RNASE4 mAbs were generated by immunizing BALB/c mice with human RNASE4 protein, followed by fusion with Sp2/0 mouse myeloma cells. A panel of 10 hybridoma clones was generated. Two clones had the highest sensitivity for RNASE4 (Clones 8A7G12 and 3F11G7) which were purified from hybridoma cell culture adapted to serum free conditions. Specificity of various mAb clones to RNASE4 protein was measured by Western Blotting and a direct ELISA method.

**RNASE4 ELISA.** ELISA plates were coated with 10 μg/ml RNASE4 mAb, 100 μl per well, in 100 mM sodium carbonate-bicarbonate buffer (pH 8.5) overnight at 4 °C. After blocking with 5 mg/ml BSA in PBS (300 μl/well) for 1 h at RT and four washes with PBST, standards and samples were added (100 μl/well) and incubated overnight at RT. After four washes with PBST, 100 μl of 0.75 μg/ml of rabbit RNASE4 pAb was added to each well and incubated for 2 h at RT. Following another four washes, alkaline phosphatase conjugated goat anti-rabbit IgG (1:1000) was added to the wells and incubated for 1 h at RT. After four washes, 100 μl of 0.5 mg/ml p-nitrophenyl phosphate in 10 mM diethanolamine, 0.5 mM $MgCl_2$, pH 9.5, was added to each well and incubated at RT for 1–4 h, and absorbance at 450 nm was measured on a plate reader. The minimum detectable dose (MDD) of human RNASE4 is typically 0.5 ng/ml. The MDD was determined by adding two standard deviations to the mean optical density value of twelve zero standard replicates and calculating the corresponding concentration. No significant cross-reactivity was observed with angiogenin (ANG, aka RNASE5) at 1 μg/ml. Cross-species reactivity was not observed with mouse RNASE4. Three samples of known concentration were tested twelve times on one plate to assess intra-assay precision. Three samples of known concentration were tested in twelve separate assays to assess inter-assay precision. Recovery of human RNASE4 spiked to levels throughout the range of the assay in human plasma was evaluated.

**Prostate tissue array.** A TMA containing 50 cases of prostate adenocarcinoma, 20 cases of BPH, and 10 cases of normal prostate tissue was purchased from US Biomax (PR807a). A core of malignant tissue marker, hepatocellular liver cancer, was included in the array. Each array spot was 1.5 mm in diameter and 5 μm in thickness. The histological diagnosis, grading using the Gleason scoring system, and TNM grading were supplied by the manufacturer for each array. Detailed information for the array can be viewed at http://www.biomax.us/tissue-arrays/Prostate/PR807a (Prostate cancer, hyperplasia and normal tissue array).

**Inhibitors and proteins.** AXL inhibitor R428 was purchased from ApexBio. MK-2206 was from Selleckchem. Wortmannin was from Sigma. Rapamycin and LY294002 were purchased from Cell Signaling Technology. His-tagged N-terminal and C-terminal AXL proteins were purchased from NovoProlabs. Recombinant human ANG was generated using a pET11 expression system in E. coli and purified by SP-Sepharose and reversed-phase HPLC as described[48]. DEPC inactivation of RNASE4 protein was performed by incubating 1 mg of RNASE4 in 1 ml PBS with 2 mM DEPC at RT for 10 min. Unreacted DEPC was quenched by incubation with 50 mM Tris-HCL, pH 7.4, at RT for 10 min. RNASE4-DEPC was desalted by G-25 Sephadex chromatography.

**RNA extraction and qRT-PCR.** Total cellular RNA was isolated using TRIzol reagent (Invitrogen) and reverse-transcribed (1 μg) to cDNA with random and oligo(dT)18 primers by M-MLV reverse transcriptase (Promega). For estimation of RNASE4 copy numbers, samples were reverse transcribed simultaneously with a standard series of cRNA samples ($10^9$–$10^3$ copies per reaction) and cDNAs amplified on a Light Cycler 480 II (Roche) using SYBER Green PCR mix (Roche). GAPDH was used as an internal control. The primers are RNASE4 forward: 5′-AGAAGCGGGTGAGAAACAA-3′, reverse: 5′-AGTAGCGATCACTGCCACCT-3′; GAPDH forward: 5′- TGAACGGGAAGCTCACTGG-3′, reverse: 5′-TCCACCACCCTGTTGCTGTA-3′.

**Cell cycle analysis.** For flow cytometry analyses, $1 \times 10^6$ cells were fixed and permeabilized using Cytofix/Cytoperm Fixation/Permeabilization Kit (BD). Cells were stained with Ki67 FITC (BD, 1:10 in BD Perm/Wash buffer), washed, and then stained with DAPI (2 μg/ml) for 10 minutes, directly prior to analysis. Cyan ADP LX7 Analyzer flow cytometer was used.

**Bioinformatics.** An in silico analysis on RNASE4 mRNA expression of Bittner Multi-cancer (n = 1911), Su Multi-cancer (n = 174) and Roth Normal 2 (n = 504) datasets in a public cancer microarray database Oncomine (http://www.oncomine.org) was performed. In Bittner Multi-cancer dataset, RNASE4 in cancer tissues were normalized to corresponding healthy tissues. In Roth Normal 2 dataset, RNASE4 in prostate tissue was compared to all other tissues to derive higher/lower RNASE4 expression in the specific normal tissue. All microarray datasets were scaled to zero by subtracting the median from each value. This step was performed by Oncomine to remove bias in signal intensity between samples. The Oncomine™ Platform (Thermo Fisher) was used for statistical analysis and visualization. Copy number gain of RNASE4 and survival probability of patients with RNASE4 copy number variations were analyzed in (DNA) TCGA Prostate dataset (n = 308) in Oncomine. RNASE4 copy numbers were analyzed in tissue specimens only, excluding blood specimens, in TCGA Prostate dataset

(n = 232). The relationship between RNASE4 copy number amplification and mRNA expression from TCGA dataset was analyzed by cBioPortal for Cancer Genomics website (http://cbioportal.org).

**IHC.** Paraffin-embedded tissue sections were deparaffinized in xylene, followed by treatment with a graded series of alcohols (100%, 95%, 70%, and 50% ethanol) and rehydration in PBS (pH 7.5). For antigen retrieval, the sections were immersed in 10 mM Sodium Citrate, 0.05% Tween 20, pH 6.0, and heated in a microwave for 20 min. After washing in PBS, endogenous peroxidases were blocked with 0.3% hydroxyl peroxide in TBS for 15 min, followed by 2 washes in TBS. The sections were blocked with 10% goat serum with 1% BSA in TBS in a humidified chamber for 2 h at RT and then incubated with the primary antibodies diluted in TBS with 1%BSA overnight at 4 °C. For RNASE4 staining, 2 μg/ml of affinity purified human RNASE4 pAb was incubated for 3 h at RT. HRP conjugates of goat anti-mouse or rabbit IgG were used as secondary antibodies. DAB was used for color development. Slides were counterstained with Modified Mayer's Hematoxylin.

**RNASE4 knockdown cell lines.** Lentiviral mediated shRNA system (pGIPZ, Open Biosystems) was used to generate control and RNASE4 knockdown cell lines. The sequences were: shControl, 5′-ATCTCGCTTGGGCGAGAGTAAGTA-3′, shRNASE4-1, 5′-ACCTGTCAGGGAGGCATT AAA-3′; shRNASE4-2, 5′-CAAAGAGATATGGAGACATAA-3′. Lentiviral particles were packaged in HEK293 cells (ATCC, Cat # CRL-1573) with the generation II packaging plasmids (psPAX2 and pMD2.G) and concentrated by Lenti-X concentrator (Clontech). Cells were infected with lentiviral particles for 24 h in the presence of Polybrene (8 μg/ml, Millipore). The medium was replaced with complete growth medium and incubated for 24 h and then selected for 4 days with 1 μg/ml puromycin. The expression of RNASE4 protein was assessed by Western blotting, and, additionally, GFP-positive cells were visualized under a fluorescence microscope.

**In vitro angiogenesis assay.** HUVEC, $1 \times 10^5$ cells in 50 μl endothelial cell basal medium (Invitrogen) were plated on a Matrigel (BD Biosciences) treated μ-slide (μ-slide angiogenesis, Ibidi) and incubated with 1 μg/ml RNASE4 and 30 μg/ml RNASE4 mAb or 30 μg/ml isotype control IgG. Tube formation was examined under microscope over a period of 4–5 h. Image analysis to quantify number of loops and total tube length was performed by ImageJ software and WimTube image analysis tool.

**Human phospho-RTK array.** Human Phospho-RTK Arrays were purchased from R&D Systems. Cells were starved overnight and treated with 1 μg/ml RNASE4 for 5 minutes before sample collection. Protein concentrations were quantified by Bradford Assay and 1.5 mg cell lysate was used per array. Pixel densities on were analyzed by ImageJ. Average background signal from each array was subtracted using PBS negative control spots during data analysis.

**Ribonucleolytic assay.** Ribonucleolytic activities of RNASE4 and RNASE4-DEPC were examined using yeast tRNA as the substrate for ANG[49]. Reactions were initiated by addition of proteins at the concentration indicated to a final volume of 300 μl system including 600 ng yeast tRNA, 0.33 M Hepes, 0.33 M NaCl, pH 7.0, and 0.1 mg/ml RNase-free BSA. After incubation at 37 °C for 120 min, 700 μl of ice-cold 3.4% perchloric acid was added and incubated on ice for 10 min, centrifuged at $14,000 \times g$ for 10 min at 4 °C, and the absorbance at 260 nm of the supernatants was measured. All buffers and water used above was filtered through Sep-Pak cartridge (Waters) to ensure that the system is RNase-free. Experiments were done in triplicates.

**Cell culture.** PC-3 (ATCC, Cat # CRL-1435), LNCaP (ATCC, Cat # CRL-1740), DU145 (Cat # HTB-81), RWPE-1 (ATCC, Cat # CRL-11609), HEK-193 (ATCC Cat # CRL-1573), and HUVEC (ATCC, Cat # CRL-1730) cells were obtained from ATCC. PC-3 and DU145 were maintained as monolayer cultures in DMEM (Cellgro; Corning) including 100 U/ml penicillin and 100 μg/ml streptomycin (Gibco; Thermo Fisher) supplemented with 10% FBS (HyClone). LNCaP was maintained as monolayer cultures in RPMI 1640 (Cellgro; Corning) including 100 U/ml penicillin and 100 μg/ml streptomycin (Gibco; Thermo Fisher) supplemented with 10% FBS (HyClone). To test the effect of DHT on RNASE4, LNCaP cells were washed in phenol red–free medium containing charcoal/dextran–treated (steroid-stripped) FBS (Sigma), incubated in this steroid-free medium for 1-2 days prior to DHT stimulation.

Human prostate epithelial cell line RWPE-1 cells were maintained in keratinocyte serum-free medium (Gibco; Thermo Fisher) supplemented with 50 μg/ml bovine pituitary extract, 5% l-glutamine, and 5 ng/ml EGF. HUVEC were maintained in Human Endothelial SFM (Gibco; Thermo Fisher) plus 5 ng/ml bFGF. Experiments with HUVEC were carried out at cell passage number 4-8.

RNASE4 monoclonal antibody secreting hybridoma cells were cultured in serum-free CD Hybridoma Medium (Gibco; Thermo Fisher) supplemented with 8 mM GlutaMAX (Thermo Fisher) for large scale Protein G affinity purification. Hybridoma cells were sequentially adapted to serum free conditions by reducing the amount of FBS (HyClone) in media in the order of 10% FBS, 5%FBS, 2.5% and

0% FBS with every passage. Antibiotics were not used in hybridoma cell cultures. All cell cultures were maintained in 5% $CO_2$ at 37 °C.

**Cell proliferation assay**. PC-3, DU145, and LNCaP cells and stable clones were plated in 24-well tissue plates in triplicate at 10,000 cells per well and returned to the incubator. Numbers of cells were counted with the aid of a Coulter Counter after trypsinization. Cells were counted daily, until cultures reached saturation. For MTT assay, PC-3, DU145 and LNCaP cells were plated ($4 \times 10^3$ cells) in quad-ruplicate for each time point in a 96-well plate. Following the attachment of all cells after 6 h, the number of viable cells was measured in the presence of indicated agents. Briefly, 10 μl of MTT (5 mg/ml) was added to the wells and incubated for 1 h. After aspiration, 100 μl of DMSO was added to each well and incubated for 5 min at 37 °C to solubilize the bio-reduced colored MTT-formazan and to lyse the cells. The optical density was read at 570 nm in a microplate reader.

**Agarose colony formation assay**. Cells were suspended in 1 ml of 10% FBS DMEM medium containing a 0.4% agarose and plated in triplicate on a firm 0.6% agarose base in 35 mm plates (10,000 cells/well) (Yoshioka et al., 2006). Cells were then placed in a 37 °C and 5% $CO_2$ incubator. Colonies of cells were allowed to grow over the course of 2 weeks. Then, 100 μl of MTT was added to each well and plates were returned to the incubator for 2 h. Number and diameter of colonies were measured under a microscope (Nikon Eclipse Ti-S). Five randomly selected fields per dish were analyzed.

**TUNEL staining**. TUNEL staining was conducted with In Situ Cell Death Detection Kit (Roche) according to manufacturer's instructions. Immunohistochemical staining of blood vessels and cell proliferation were determined by CD31 (1:200, Abcam) and Ki-67 (1:500, Santa Cruz) antibodies. Slides were examined by microscope (Nikon Eclipse Ti-S). Two independent scientists, who were blind to clinical data, independently scored average RNASE4 expression in human tissue arrays as absent (0 or -), low (1 or +), moderate (2 or ++), high (3 or +++) or very high (4 or ++++).

**Statistics and reproducibility**. Sample size and number of replicates for each experiment was described in figure legends. Data were presented as means ± SEM. GraphPad Prism was used for statistical analysis. Comparisons between 2 groups were analyzed by 2-tailed Student's $t$ test and comparisons of multiple groups were done by 1-way ANOVA, post hoc intergroup comparisons, and Tukey's test. Kaplan-Meier survival curves were analyzed using log-rank test. Receiver operating characteristic (ROC) curves were generated by MedCalc and Prism. Regression analyses were performed by MedCalc and Stata Software. The p values were indicated by stars as follows: $*p < 0.05$, $**p < 0.01$, $***p < 0.001$.

**Reporting summary**. Further information on research design is available in the Nature Research Reporting Summary linked to this article.

## Data availability

No datasets were generated or analyzed during the current study. All data generated or analyzed during this study are included in this published article and in its Supplementary Information files. Source data used to generate graphs and charts are included as Supplementary Data 1 and 2. Unedited gel images are included in Supplementary Figs. 13–16.

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

## Acknowledgements

We thank Johns Hopkins University for supplying patient plasma samples and clinical information. This research was supported by NIH grants R01CA105241 and R01HL135160 (to G.-F.H.), Tufts Collaborative Cancer Biology Award (to N.V and G.-F.H.), and Department of Defense Prostate Cancer Research Program, Award No W81XWH-14-2-0182, W81XWH-14-2-0183, W81XWH-14-2-0185, W81XWH-14-2-0186, and W81XWH-15-2-0062 to Prostate Cancer Biorepository Network (PCBN).

## Author contributions

N.V. and G.-F.H. developed the hypothesis, designed experiments, analyzed the data, and wrote the manuscript. N.V., J.S., S.L., Z.X., and G.-F.H. performed experiments. All authors reviewed and edited the manuscript.

## Competing interests

The authors declare no competing interests.
