## [Peer Review File · Communications Biology]

Reviewers' comments:

Reviewer #1 (Remarks to the Author):

Hu and coworkers report on the novel finding that human RNase 4 is a serum biomarker for prostate cancer and that levels of RNase 4 correlate with disease progression. To do so, the authors compared RNase 4 levels to those PSA and angiogenin, which have both been reported previously to be serum biomarkers of prostate cancer. In general, RNase 4 was as useful or better than the other two markers. Their assertion that RNase 4 activates AKT and S6 is supported well by immunoblots and assays with inhibitors. Their identification of AXL as the interacting RTK is likewise supported well. Finally, the value of an RNase 4-specific mAb to inhibit prostate cancer growth is an interesting, though unsurprising, result given the results knockdown experiments, that it is a competitive binder, and the high amount of antibody used in the experiments (30–90× the amount of RNase 4 used in similar cell culture experiments and at the upper end of the amount of antibody-based drugs given to patients in mouse experiments, assuming 80-kg individuals). Regardless, the mAb data support the authors' claims with the RNase 4 knockdown experiments on the importance of RNase 4 for prostate tumor growth. The manuscript is well written with few errors, and the figures are relevant and clear.

Comments:

1. Given the literature and their few experiments, the authors need to exert a bit more care in making claims on the importance of the angiogenic ability of RNase 4. In particular, the authors' claim that RNase 4 induces tumor angiogenesis is not fully supported because the authors used only tube-formation experiments, which make use of basal media (no serum or growth factors). Moreover, ref. 23 (by the same author group) already showed that RNase 4 can stimulate angiogenesis, and the additional experiments here (i.e., blocking RNase 4 with an antibody) do not make the case that RNase 4 is important for angiogenesis in prostate cancer specifically or even angiogenesis in general. Compared to RNase 4, angiogenin is a more potent stimulator of angiogenesis that is expressed at higher levels and is also often upregulated in prostate cancer (as shown by the authors), so angiogenin could be the cause of the tumor angiogenesis. For this reason, the text (e.g., lines 302–306) requires some revision.
2. Lines 45–49: RNase 4 does not have the "strictest" substrate specificity of the family. ANG is likely more specific for its biological substrate, pRNA. The authors should replace "substrate" with "nucleotide".
3. Lines 89–95 and Supplementary Figure 2d: For PSA levels of <2 ng/mL, the AUC = 0.72 but the confidence interval is 0.41–1.00 (where 0.5 is a random predictor). Thus, the claim in lines 94–95 does not appear to be significant.
4. Supplementary Figure 2: The authors should insert p values in panels f–i (as in Supplementary Figure 3 d–g) and all graphs in Figure 2.
5. Supplementary Figure 3: Why does angiogenin do better than RNase 4 at PSA levels of <2 ng/mL, and why does angiogenin at PSA levels of <4 ng/mL do better than the general ANG case?
6. Line 191: RNase 4 seems to translocate the cell membrane (ref. 23) and could have intracellular targets, but only RTKs (extracellular targets) were considered by the authors.
7. Figure 4c: This immunoblot gives little confidence in the conclusions. Why is there a second band here and not in Figure 4b? In general, how many replicate blots were done? The experiment should be repeated or, at least, error bars should be added to the quantification in the accompanying graph.
8. Figure 4e: The quantification for p-S6/S6 does not match the blot (specifically, the +R428 conditions give no band on the blot but are higher than the DMSO-only condition on the quantification graph).
9. Lines 238–239: Cell proliferation and angiogenesis by RNase 4 might or might not have the same mechanism. Also, why use bovine FGF as an angiogenesis control instead of human ANG, which is homologous to RNase 4 and would thus better demonstrate specificity?
10. RNase 4 has ribonucleolytic activity. An important issue that the authors did not address is whether or not their data are dependent on ribonucleolytic activity. This issue is a "hole" in the manuscript that could be addressed by using a wounded RNase 4 variant.
11. Methods: The authors need to be more specific in many instances. For example, how were RNase A and angiogenin (human?) obtained or made? Is RNase 4 always added at 1 µg/mL? How were the MTT assays performed?

Reviewer #2 (Remarks to the Author):

In the manuscript "Ribonuclease 4 is associated with aggressiveness and progression of prostate cancer" by Vanli et. al., the authors describe the use of the RNase 4 as a novel biomarker for prostate cancer. The authors report on the utility of RNase 4 to develop diagnostic and therapeutic strategies based on this nuclease. The manuscript is well designed, provides clear description of the results and it is an innovative study. This contribution introduces an impactful role of the nuclease RNase 4 in prostate cancer diagnosis and treatment. Before considering this contribution for publication *Communication Biology*, some important issues need to be addressed by the authors.

Major comments.

The diagnostic results of RNase4 and PSA are not compared and described in a fair manner. This comparison is critical, directly impacting on the claims of the paper, because PSA is still a relevant biomarker for the diagnosis of prostate cancer. So, I suggest the followings:

1. Move the information from supplementary figure 2a and 2b to Figure 1. In this way the reader could directly visualize and compare both biomarkers simultaneously, in the main text.

2. The PSA results are superior (Sensitivity 95%, Specificity 99%) to the results reported for RNAase 4 (Sensitivity 94%, Specificity 80%). Please clearly address and discuss this data in the results and discussion sections.

3. Pearson analysis between PSA and RNase4 is missing. Please include this analysis as supplementary information and describe accordingly in the results section.

4. Page 5, Lines 92-94 the authors have claimed the following: "A PSA value of 4 ng/ml and above is considered suspicious for the presence of prostate cancer, but for patients with PSA results below 2 ng/ml, there are currently no available biomarkers". This sentence should be referenced.

5. Page 5, Lines 94-95, the following is claimed by the authors: "These results suggest that RNASE4 performs better than PSA at predicting earlier disease states in patients with PSA levels less than 2 ng/ml". To be thorough, please indicate the number of patients with less than 2ng/mL in the PSA test that have been detected with RNAase 4.

6. The method section is incomplete. I believe that the subsection of cell culture is missing or described partially elsewhere. Please double check that all the methods and reagents are included in the method section.

Minor comments:

7. Please use RNase 4 instead of RNASE4. (Q Rev Biophys. 2011 Feb; 44(1): 1-93.)

8. Fig S12, typo mistake in the units (5 ng/m)

Reviewer #3 (Remarks to the Author):

In this manuscript encoded "Ribonuclease 4 is associated with aggressiveness and progression of prostate cancer" Vanli and colleagues investigated the clinical value of RNase4 expression as biomarker of prostate carcinogenesis and prostate cancer progression. According to presented data, plasma RNase4 level is elevated in prostate cancer and is positively correlated to disease stage. Moreover, authors provide evidence that RNase 4 induces AXL activation that in turn, promotes AKT/S6K activation and tumor cells proliferation. They also show that RNase 4 inhibition by a monoclonal antibody inhibits the growth of xenograft human prostate tumors.

Overall is a very well written manuscript with a nice experimental design and novel data of clinical significance.

Minor comments:

1. The part of the manuscript that presents the RNase4 driven activation of AXL is somehow descriptive and the related mechanism has not been studied in depth.
2. Supplementary figure 6a: Is the RNase4 copy number amplification positively correlated with its expression?
3. Prostate adenocarcinoma TCGA data: Is RNase4 expression correlated with stage and grade?

Point-by-point response

We thank three reviewers very much for their insightful comments and constructive suggestions. We have performed additional experiments and have revised the manuscript incorporating new results and addressed issues raised by the reviewers.

New figure panels have been added in the revised submission including Fig. 1f, 5e, 5f, 5g and Supplementary Fig. 6b, 6c.

Supplementary Fig. 2a, 2b, and 2c in the original submission is now Fig. 1c, 1d, and 1e, respectively. Fig. 4c, 4f, Supplementary Fig. 2d, 2e, 3h, 3i have been deleted.

Reviewer #1 (Remarks to the Author):

Hu and coworkers report on the novel finding that human RNase 4 is a serum biomarker for prostate cancer and that levels of RNase 4 correlate with disease progression. To do so, the authors compared RNase 4 levels to those PSA and angiogenin, which have both been reported previously to be serum biomarkers of prostate cancer. In general, RNase 4 was as useful or better than the other two markers. Their assertion that RNase 4 activates AKT and S6 is supported well by immunoblots and assays with inhibitors. Their identification of AXL as the interacting RTK is likewise supported well. Finally, the value of an RNase 4-specific mAb to inhibit prostate cancer growth is an interesting, though unsurprising, result given the results knockdown experiments, that it is a competitive binder, and the high amount of antibody used in the experiments (30–90× the amount of RNase 4 used in similar cell culture experiments and at the upper end of the amount of antibody-based drugs given to patients in mouse experiments, assuming 80-kg individuals). Regardless, the mAb data support the authors' claims with the RNase 4 knockdown experiments on the importance of RNase 4 for prostate tumor growth. The manuscript is well written with few errors, and the figures are relevant and clear.

Response: We thank the reviewer for the positive comments and overall enthusiasm to the manuscript.

Comments:

1. Given the literature and their few experiments, the authors need to exert a bit more care in making claims on the importance of the angiogenic ability of RNase 4. In particular, the authors' claim that RNase 4 induces tumor angiogenesis is not fully supported because the authors used only tube-formation experiments, which make use of basal media (no serum or growth factors). Moreover, ref. 23 (by the same author group) already showed that RNase 4 can stimulate angiogenesis, and the additional experiments here (i.e., blocking RNase 4 with an antibody) do not make the case that RNase 4 is important for angiogenesis in prostate cancer specifically or even angiogenesis in general. Compared to RNase 4, angiogenin is a more potent stimulator of angiogenesis that is expressed at higher levels and is also often upregulated in prostate cancer (as shown by the authors), so angiogenin could be the cause of the tumor angiogenesis. For this reason, the text (e.g., lines 302–306) requires some revision.

Response: To downplay a possible role of RNASE4 in tumor angiogenesis, we have deleted the statement that “This is the first time that RNASE4 is found to have a direct role toward cancer cells. RNASE4 is a new addition to a group of angiogenic proteins including ANG, VEGF, bFGF, and EGF⁴⁵⁻⁴⁸ that simultaneously stimulate cancer cell progression and induce tumor angiogenesis, thus playing a dual role in promoting cancer progression”. The revised text reads: “We have reported previously that RNASE4 was able to stimulate angiogenesis in an *in vitro* assay²³. Now we show that RNASE4 mAbs inhibited RNASE4-induced angiogenesis *in vitro* (Fig. 6c and Supplementary Fig. 11) and decreased neovessel density in the tumor tissue (Fig. 7d and 8c). These data suggest a role of RNASE4 in angiogenesis, a function similar to that of ANG³¹” (page 15, lines 1-4).

2. Lines 45–49: RNase 4 does not have the “strictest” substrate specificity of the family. ANG is likely more specific for its biological substrate, pRNA. The authors should replace “substrate” with “nucleotide”.

Response: The word “substrate” has been replaced with “nucleotide” (page 3, line 3 from bottom).

3. Lines 89–95 and Supplementary Figure 2d: For PSA levels of <2 ng/mL, the AUC = 0.72 but the confidence interval is 0.41–1.00 (where 0.5 is a random predictor). Thus, the claim in lines 94–95 does not appear to be significant.

Response: We have deleted Supplementary Fig. 2d and 2e, and the corresponding description in the manuscript text.

4. Supplementary Figure 2: The authors should insert p values in panels f–i (as in Supplementary Figure 3 d–g) and all graphs in Figure 2.

Response: p values have been inserted in panels f–i of Supplementary Fig. 2 (panels a–d in the revised Supplementary Fig. 2).

5. Supplementary Figure 3: Why does angiogenin do better than RNase 4 at PSA levels of <2 ng/mL, and why does angiogenin at PSA levels of <4 ng/mL do better than the general ANG case?

Response: The reasons for the different sensitivity of ANG in patients with different PSA levels are unknown. We have deleted panels h and i of Supplementary Fig. 3, so the revised manuscript no longer discusses the diagnosis performance of RNase 4 or ANG in patients with PSA values <2 or 4 ng/mL. This modification is also consistent with the revision made in Supplementary Fig. 2 where we have deleted the previous panels d and e (please see response to Comment 3 above).

6. Line 191: RNase 4 seems to translocate the cell membrane (ref. 23) and could have intracellular targets, but only RTKs (extracellular targets) were considered by the authors.

Response: Since RNase 4 is a secreted protein and since it activates the PI3K-AKT-mTOR pathway as shown in Fig. 3 and supplementary Fig. 8, a cell surface receptor is likely for RNase4. We have added this rationale to page 10, lines 1–2, where it reads “Since RNase4 is a secreted protein, the above findings suggest a receptor-mediated function of RNase4. We performed...”.

7. Figure 4c: This immunoblot gives little confidence in the conclusions. Why is there a second band here and not in Figure 4b? In general, how many replicate blots were done? The experiment should be repeated or, at least, error bars should be added to the quantification in the accompanying graph.

Response: The second band of phosphorylated Axl in Fig. 4c was more obvious than in Fig. 4b because the two bands were more separated in Fig. 4c as indicated by the two bands in the total Axl blots. As recommended by the reviewer, immunoblot alone did not allow us to conclude that ANG and RNase A did not stimulate Axl phosphorylation so we have deleted Fig. 4c in the revised manuscript. In general, immunoblotting experiments were repeated at least once. Data shown were blots and quantification of a representative experiment. We have made this clearer in figure legends.

8. Figure 4e: The quantification for p-S6/S6 does not match the blot (specifically, the +R428 conditions give no band on the blot but are higher than the DMSO-only condition on the quantification graph).

Response: Figure 4e (Figure 4d in the revised manuscript) has been re-quantified and the quantification matches the blot now.

9. Lines 238–239: Cell proliferation and angiogenesis by RNase 4 might or might not have the same mechanism. Also, why use bovine FGF as an angiogenesis control instead of human ANG, which is homologous to RNase 4 and would thus better demonstrate specificity?

Response: Basic FGF (bFGF) has a more robust angiogenesis-inducing activity than ANG in *in vitro* experiment. We have made it clearer in the revised manuscript (page 12, lines 4-5) where it reads “.....but had no effect to endothelial cell tube formation induced by basic fibroblast growth factor (bFGF), an unrelated angiogenic factor (Supplementary Fig. 12)”. As suggested by the reviewer that the RNASE 4 may induced cell proliferation and angiogenesis by a different mechanism, so we made it clearer that these experiments demonstrate the activity of RNASE 4 could be inhibited by mAb without mechanistic implication (page 12, lines 5-8) where it reads “It is currently unclear if RNASE4 induces cell proliferation and angiogenesis by the same mechanism, but these results demonstrate the effectiveness and specificity of RNASE4 mAb in inhibiting the activity of RNASE4 in these cellular events”.

10. RNase 4 has ribonucleolytic activity. An important issue that the authors did not address is whether or not their data are dependent on ribonucleolytic activity. This issue is a “hole” in the manuscript that could be addressed by using a wounded RNase 4 variant.

Response: We have performed a new experiment to show that DEPC-treated RNASE4 failed to rescue the effect of *RNASE4* shRNA knockdown. These results indicate that the ribonucleolytic activity of RNASE4 is essential for cell proliferation. They are included as Fig. 5e, 5f, and 5g in the revised manuscript and described in the relevant part of the manuscript text (page 11, lines 1-6) where it reads “Next, we examined if ribonucleolytic activity of RNASE4 was essential to induce cell proliferation. For this purpose, RNASE4 protein was treated with 2 mM of diethyl pyrocarbonate (DEPC) in PBS for 10 min, a method known to inactivate RNase by modifying His and Lys residues in the catalytic site⁴³. DEPC-treated RNASE4 (Fig. 5e) had no enzymatic activity using yeast tRNA as the substrate (Fig. 5f), and was not able to rescue the effect of RNASE4 knockdown in cell proliferation (Fig. 5g). These results indicate that the ribonucleolytic activity is essential for RNASE4 to mediate cell proliferation”.

11. Methods: The authors need to be more specific in many instances. For example, how were RNase A and angiogenin (human?) obtained or made? Is RNase 4 always added at 1 µg/mL? How were the MTT assays performed?

Response: We have added new sections to the Methods including Ribonucleolytic assay, Cell culture, Cell proliferation assay, Agarose-colony formation assay, and TUNEL staining (pages 21-24). We have also added more detailed descriptions to sections of Recombinant human RNASE4 proteins and antibodies (page 17), Inhibitors and proteins (pages 18-19), and RNASE5 knockdown cell lines (page 21).

Reviewer #2 (Remarks to the Author):

In the manuscript "Ribonuclease 4 is associated with aggressiveness and progression of prostate cancer" by Vanli et. al., the authors describe the use of the RNase 4 as a novel biomarker for prostate cancer. The authors report on the utility of RNase 4 to develop diagnostic and therapeutic strategies based on this nuclease. The manuscript is well designed, provides clear description of the results and it is an innovative study. This contribution introduces an impactful role of the nuclease RNase 4 in prostate cancer diagnosis and treatment. Before considering this contribution for publication Communication Biology, some important issues need to be addressed by the authors.

Response: We thank the reviewer for the positive comments and overall enthusiasm to the manuscript.

Major comments.

The diagnostic results of RNase4 and PSA are not compared and described in a fair manner. This comparison is critical, directly impacting on the claims of the paper, because PSA is still a relevant biomarker for the diagnosis of prostate cancer. So, I suggest the followings:

1. Move the information from supplementary figure 2a and 2b to Figure 1. In this way the reader could directly visualize and compare both biomarkers simultaneously, in the main text.

Response: We have moved Supplementary Fig. 2a, 2b, and 2c to Figure 1 as Fig. 1c, 1d, and 1e in the revised manuscript.

2. The PSA results are superior (Sensitivity 95%, Specificity 99%) to the results reported for RNAase 4 (Sensitivity 94%, Specificity 80%). Please clearly address and discuss this data in the results and discussion sections.

Response: We have stated in the Result section that “Thus, as a single blood marker for diagnosis of prostate cancer, PSA remains as a superior marker over RNASE4 (page 5, lines 10-11). We have also discussed in the discussion section on page 13, lines 1-2 from bottom that “While PSA is superior to RNASE4 when used as an individual serum marker for prostate cancer, RNASE4 has the potential to distinguish cancer from benign growth”.

3. Pearson analysis between PSA and RNase4 is missing. Please include this analysis as supplementary information and describe accordingly in the results section.

Response: Pearson analysis between PSA and RNASE4 has been added in the revised manuscript as Fig. 2f and discussed on page 5, lines 6-7 from bottom where it reads “Further, a positive correlation (Pearson $r = 0.27$, $p < 0.0031$) was found between PSA and RANSE4 in this cohort of prostate cancer patients (Fig. 1f)”.

4. Page 5, Lines 92-94 the authors have claimed the following: “A PSA value of 4 ng/ml and above is considered suspicious for the presence of prostate cancer, but for patients with PSA results below 2 ng/ml, there are currently no available biomarkers”. This sentence should be referenced.

Response: This sentence has been deleted in the revised manuscript in response to Comment #3 of Reviewer 1.

5. Page 5, Lines 94-95, the following is claimed by the authors: “These results suggest that RNASE4 performs better than PSA at predicting earlier disease states in patients with PSA levels less than 2 ng/ml”. To be thorough, please indicate the number of patients with less than 2ng/mL in the PSA test that have been detected with RNAase 4.

Response: This sentence has also been deleted in the revised manuscript in response to Comment #3 of Reviewer 1.

6. The method section is incomplete. I believe that the subsection of cell culture is missing or described partially elsewhere. Please double check that all the methods and reagents are included in the method section.

Response: The method section has been expended to cover all the methods that were used in the study. Please see response to Comment #11 of Reviewer 1.

Minor comments:

7. Please use RNase 4 instead of RNASE4. (Q Rev Biophys. 2011 Feb; 44(1): 1–93.)

Response: We appreciate the suggestion of the reviewer. RNase is indeed the customary abbreviation of ribonuclease. We have changed RNASE to RNase in the revised manuscript when it refers to ribonuclease in general (page 3, line 16; page 5, line 5 from bottom; page 11, line 2; page 21, line 2 from bottom; page 22, line 3). However, RNASE4 is the official name of ribonuclease 4 recommended by

HUGO gene Nomenclature Committee (HGNC), we hope the reviewer will agree that RNASE4 is a more appropriate name when it refers ribonuclease 4.

8. Fig S12, typo mistake in the units (5 ng/m)

Response: The typo has been corrected (5 ng/ml).

Reviewer #3 (Remarks to the Author):

In this manuscript encoded “Ribonuclease 4 is associated with aggressiveness and progression of prostate cancer” Vanli and colleagues investigated the clinical value of RNase4 expression as biomarker of prostate carcinogenesis and prostate cancer progression. According to presented data, plasma RNase4 level is elevated in prostate cancer and is positively correlated to disease stage. Moreover, authors provide evidence that RNase 4 induces AXL activation that in turn, promotes AKT/S6K activation and tumor cells proliferation. They also show that RNase 4 inhibition by a monoclonal antibody inhibits the growth of xenograft human prostate tumors.

Overall is a very well written manuscript with a nice experimental design and novel data of clinical significance.

Response: We thank the reviewer for the positive comments and overall enthusiasm to the manuscript.

Minor comments:

1. The part of the manuscript that presents the RNase4 driven activation of AXL is somehow descriptive and the related mechanism has not been studied in depth.

Response: We understand that the mechanistic insight in the relationship between RNASE4 and AXL activation is limited at present. To avoid over-interpretation of the available data, we have deleted the proposed model in Fig. 4f, and have modified the text to indicate that the results only “suggest a relationship between RNASE4 and AXL in regulating AKT and S6 phosphorylation in prostate cancer cells” (page 10, lines 12-13).

2. Supplementary figure 6a: Is the RNase4 copy number amplification positively correlated with its expression?

Response: The *RNASE4* copy number is positively correlated with the mRNA level. A new figure panel of Pearson analysis is included as Supplementary Fig. 6b in the revised manuscript, and described on page 7, lines 1-2.

3. Prostate adenocarcinoma TCGA data: Is RNase4 expression correlated with stage and grade?

Response: No correlation was found between *RNASE4* mRNA levels and Gleason scores. A new figure panel has been included as Supplementary Fig. 6c in the revised manuscript and described on page 7, lines 3-4.

REVIEWERS' COMMENTS:

Reviewer #1 (Remarks to the Author):

The authors have made excellent revisions to their manuscript, which is now suitable for publication.

I did identify one typographical error: on page 15 there should be an "and" between "...Supplementary Fig. 11)" and "decreased..." (There is in the rebuttal statement but not in the revised manuscript itself.)

Reviewer #2 (Remarks to the Author):

The authors have addressed accordingly all my comments. I recommend this contribution for publication in the present form.

Reviewer #3 (Remarks to the Author):

The revisions have addressed my comments and I consider that the revised manuscript is significantly improved